# Methylation of RNA polymerase II non-consensus Lysine residues marks early transcription in mammalian cells

João D Dias[1,2,3], Tiago Rito[1†], Elena Torlai Triglia[1†], Alexander Kukalev[1], Carmelo Ferrai[1,2], Mita Chotalia[2], Emily Brookes[2‡], Hiroshi Kimura[4*], Ana Pombo[1,2*]

[1] Epigenetic Regulation and Chromatin Architecture Group, Berlin Institute for Medical Systems Biology, Max-Delbrück Centre for Molecular Medicine, Berlin, Germany; [2] Genome Function Group, MRC Clinical Sciences Centre, Imperial College London, London, United Kingdom; [3] Graduate Program in Areas of Basic and Applied Biology, University of Porto, Porto, Portugal; [4] Graduate School of Bioscience and Biotechnology, Tokyo Institute of Technology, Yokohama, Japan

**Abstract** Dynamic post-translational modification of RNA polymerase II (RNAPII) coordinates the co-transcriptional recruitment of enzymatic complexes that regulate chromatin states and processing of nascent RNA. Extensive phosphorylation of serine residues at the largest RNAPII subunit occurs at its structurally-disordered C-terminal domain (CTD), which is composed of multiple heptapeptide repeats with consensus sequence $Y_1$-$S_2$-$P_3$-$T_4$-$S_5$-$P_6$-$S_7$. Serine-5 and Serine-7 phosphorylation mark transcription initiation, whereas Serine-2 phosphorylation coincides with productive elongation. In vertebrates, the CTD has eight non-canonical substitutions of Serine-7 into Lysine-7, which can be acetylated (K7ac). Here, we describe mono- and di-methylation of CTD Lysine-7 residues (K7me1 and K7me2). K7me1 and K7me2 are observed during the earliest transcription stages and precede or accompany Serine-5 and Serine-7 phosphorylation. In contrast, K7ac is associated with RNAPII elongation, Serine-2 phosphorylation and mRNA expression. We identify an unexpected balance between RNAPII K7 methylation and acetylation at gene promoters, which fine-tunes gene expression levels.

*For correspondence: hkimura@
bio.titech.ac.jp (HK); ana.pombo@
mdc-berlin.de (AP)

† These authors contributed equally to this work

Present address: ‡ MRC Laboratory for Molecular Cell Biology, University College London, London, United Kingdom

## Introduction

Transcription of protein-coding genes is a complex process involving a sequence of RNA processing events that occur at different stages of the transcription cycle. Co-transcriptional recruitment of chromatin modifiers and RNA processing machinery is modulated through a complex array of post-translational modifications at the C-terminal domain (CTD) of RPB1, the largest subunit of RNAPII. This unique domain constitutes a docking platform for protein complexes that cap, splice and polya-denylate newly-made RNAs (*Bentley, 2014*; *Buratowski, 2009*; *Egloff et al., 2012*; *Eick and Geyer, 2013*; *Hsin and Manley, 2012*). The CTD also integrates signaling cascades that, for example, coordinate the DNA damage response and chromatin remodeling with gene expression (*Munoz et al., 2009*; *Winsor et al., 2013*).

The CTD is a large, structurally disordered domain composed of a tandem heptapeptide repeat structure with the canonical sequence $Y_1$-$S_2$-$P_3$-$T_4$-$S_5$-$P_6$-$S_7$. Extensive remodeling of the CTD occurs during distinct steps of the transcription cycle (*Buratowski, 2003*, *2009*). RNAPII binds to promoter regions in a hypophosphorylated state, before the CTD becomes phosphorylated at Serine-5 (S5p) and Serine-7 (S7p), marking the earliest stages of transcription (*Akhtar et al., 2009*;

**eLife digest** Genes are sections of DNA that encode the instructions to make proteins. When a gene is switched on, the section of DNA is copied to make molecules of messenger ribonucleic acid (RNA) in a process called transcription. These messenger RNAs are then used as templates for protein production. In animals, plants and other eukaryotic organisms, an enzyme called RNA polymerase II is responsible for making messenger RNA molecules during transcription. This enzyme is made up of several proteins, the largest of which contains a long tail, called the carboxy-terminal domain. This domain is crucial for transcription because it serves as a landing platform for other enzymes that help to make the RNA molecules.

The carboxy-terminal domain contains multiple repeats of a string of seven amino acids (the building blocks of proteins). Normally, each repeat contains three amino acids called serines. However, in humans and other mammals, one of these serines is often substituted with another amino acid called lysine instead. This lysine (referred to as Lysine-7) was known to be modified by the addition of a chemical group called an 'acetyl' tag, but it was not clear how this tag affected transcription.

Dias, Rito, Torlai Triglia et al. carried out an in-depth study into how Lysine-7 is modified in mouse cells, and what effects these modifications have on transcription. The experiments show that Lysine-7 can also be modified by the addition of a different chemical group, called a 'methyl' tag. This new modification is also found in flies, worms and human cells, which suggests that it is generally important for transcription.

Next, Dias, Rito, Torlai Triglia et al. found that in mouse stem cells, methyl tags are added to Lysine-7 during the earliest steps of transcription, before the acetyl tags are added. Further experiments show that a balance between the addition of methyl tags and acetyl tags to Lysine-7 fine-tunes the activity of RNA polymerase II. These findings add to our understanding of how cells control the activity of RNA polymerase II at different genes. Future challenges are to find out which enzymes are responsible for adding and removing these chemical tags, and how the balance between the methyl and acetyl modifications is controlled.

*Chapman et al., 2007*; *Tietjen et al., 2010*). Productive elongation is characterized by an increase in phosphorylation of Serine-2 (S2p) throughout gene bodies, with the highest levels found around transcription end sites (TES). S5p is important for recruitment of the capping machinery, while S2p is involved in the recruitment of splicing and polyadenylation factors (*Corden, 2013*; *Ghosh et al., 2011*; *Gu et al., 2013*; *Lunde et al., 2010*).

Although the tandem repeat structure of the CTD was acquired very early in eukaryotic evolution, and general features of serine phosphorylation are fairly conserved from yeast to mammals, the number of repeats is highly variable among different taxa (*Chapman et al., 2008*; *Yang and Stiller, 2014*). The most complex multicellular organisms, such as vertebrates, generally have longer CTDs (e.g. 52 heptad repeats in mammals), whereas *Drosophila melanogaster*, *Caenorhabditis elegans* and unicellular yeast have 44, 42 and 26–29 copies, respectively. The mammalian CTD retains a core of 21 consensus repeats, but has accumulated a diversity of non-consensus repeats, particularly at its most C-terminal region (*Figure 1a*). In vertebrates, non-canonical amino-acid residues occur most frequently at the seventh position of the heptapeptide repeat, and the most frequent substitution replaces the canonical S7 residue with a lysine (K7; *Figure 1a*). The number of non-canonical K7-containing repeats increases from zero in yeast to one, three and eight repeats in *C. elegans, D. melanogaster* and vertebrates, respectively (*Figure 1b*). Previous work has shown that non-canonical CTD-K7 residues can be acetylated, and that CTD-K7ac is associated with transcriptional pausing at epidermal growth factor (EGF)-inducible genes in mouse fibroblasts (*Schroeder et al., 2013*). Evolutionary analyses also suggest that CTD-K7ac played a role in the origin of complex Metazoan lineages (*Simonti et al., 2015*).

To further explore the increasing complexity of CTD modifications over evolution, their temporal sequence, and how they interplay with each other, we have investigated the possibility of additional modification of non-canonical CTD residues. We identify mono- and di-methylation of CTD-K7

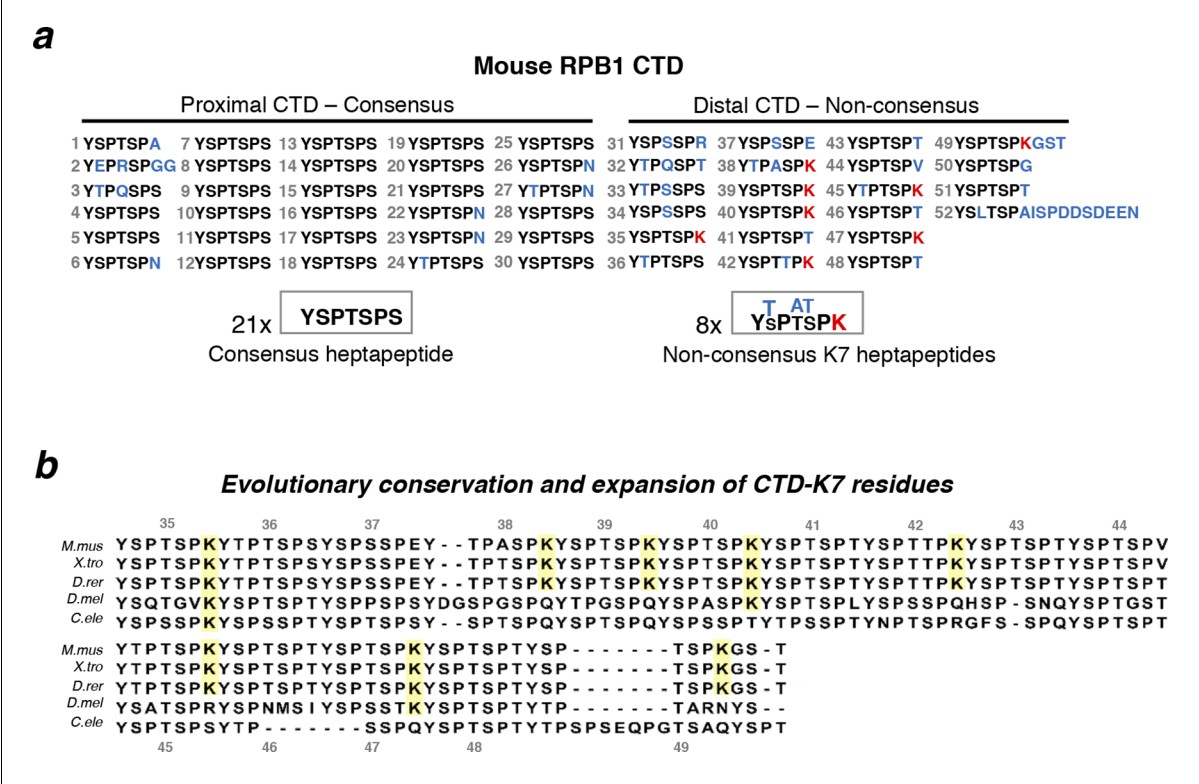

**Figure 1.** Structure and evolutionary conservation of the C-terminal domain of RPB1. (a) Mouse RPB1 CTD is composed of 52 heptapeptide repeats with consensus amino-acid sequence YSPTSPS, which is represented 21 times at the most proximal CTD region. Non-consensus amino acids are enriched for at the distal region. Most abundant non-consensus residues are lysines, all found at heptad position 7 (K7; represented in red). Other non-consensus residues are represented in blue. (b) Amino-acid sequence alignment of the most distal part of the CTD containing K7 residues across different species: *Mus musculus (M. mus); Xenopus tropicalis (X. tro); Danio rerio (D. rer); Drosophila melanogaster (D. mel)*; and, *Caenorhabditis elegans (C. ele)*. Conservation of CTD K7 residues is highlighted in yellow. CTD repeat numbering was done according to the mouse CTD sequence between repeats 35 and 49, and aligned to the other species CTDs from the position of the repeat containing the first lysine in each organism. RPB1, RNA polymerase II large subunit; CTD, C-terminal domain.

residues in both vertebrates and invertebrates. We produce new antibodies specific to CTD-K7me1 and CTD-K7me2 and show that these novel modifications precede or accompany phosphorylation of S5 and S7, upstream of S2 phosphorylation. Using biochemical and genome-wide approaches, we show that CTD-K7 methylation is present at the promoters of genes that are productively transcribed into mature RNA, but defines the earliest stages of the transcription cycle. Through detailed analysis of abundance and distribution of different CTD modifications at gene promoters in embryonic stem (ES) cells, we show that gene expression levels depend on the balance between CTD-K7 methylation and acetylation.

## Results

### Mutation of CTD-K7 residues is compatible with cell viability

To study the importance of non-consensus CTD-K7 residues on cell viability and their potential for post-translational methylation, we generated stable mouse NIH-3T3 cell lines expressing α-amanitin-resistant RPB1 bearing CTD-K7 mutations (*Figure 2a*). In this system, the endogenous α-amanitin-sensitive RPB1 is continually depleted and functionally replaced by the resistant variant (*Nguyen et al., 1996*). CTD-K7 residues were mutated into serine (S7) residues to restore the consensus sequence of the CTD heptapeptide. We avoided the more traditional lysine to arginine substitution, as a non-canonical arginine residue is present at the CTD in position 7 of repeat 31 and

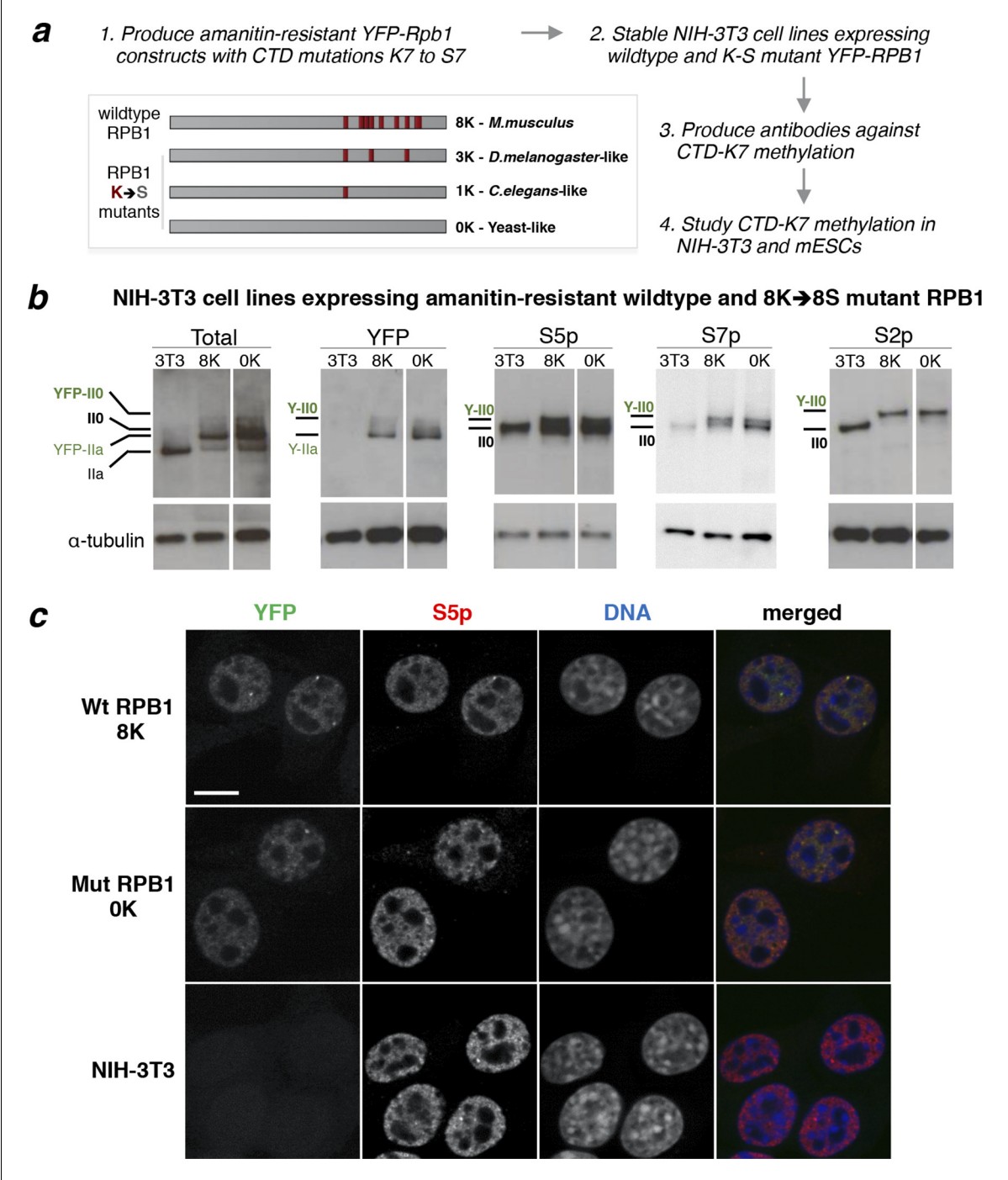

**Figure 2.** Mutation of CTD-K7 to -S7 residues does not interfere with RPB1 stability, phosphorylation or subcellular localization. (a) Outline of strategy used to generate mouse cell lines bearing K7-to-S7 mutations and study CTD-K7 methylation. Red bars represent CTD repeats with K7 residues. The nomenclature of the cell lines is indicative of the number of K7 residues retained in each α-amanitin-resistant Rpb1 constructs. (b) Expression and phosphorylation levels of RPB1 in cell lines expressing wild-type and mutant YFP-Rpb1 construct. Levels of total RPB1, YFP, S5p, S7p and S2p were analyzed by western blotting in total cell extracts from NIH-3T3 (3T3) and from NIH-3T3 cell lines expressing wild-type (8K) and mutant (0K) RPB1. Hypo-(IIa) and hyperphosphorylated (II0) isoforms of YFP-Rpb1 constructs migrate slower than wild-type construct detected in 3T3 due to the YFP tag (Y-IIa and Y-II0 respectively). Total RPB1 was detected with an antibody to the N-terminus of RPB1. α-tubulin was used as loading control. For each blot/ antibody, samples were run in the same gel, and re-ordered to improve clarity. Complete western blots are shown in *Figure 2—figure supplement 1*. (c) Whole-cell detection of RPB1 expression in wild-type (8K), mutant (0K) and untransfected NIH-3T3 fibroblasts. Expression and distribution of total RNAPII (YFP, green) is similar in 8K and 0K cell lines. Immunofluorescence of S5p (pseudo-colored red) shows similar pattern and distribution in the

*Figure 2 continued on next page*

*Figure 2 continued*

three cell lines. DNA (pseudo-colored blue) was counterstained with TOTO-3. Scale bar, 10 μm. CTD, C-terminal domain; RPB1, RNA polymerase II large subunit; YFP, yellow fluorescent protein.

The following figure supplement is available for figure 2:

**Figure supplement 1.** Complete western blots used in *Figure 2a*.

undergoes methylation *in vivo* (*Sims et al., 2011*). Therefore, artificial expansion of R7 residues in the CTD could confound our investigation of CTD-K7 methylation.

To explore the effect of the number and position of different K7 residues in the mouse CTD, we generated α-amanitin-resistant *YFP-Rpb1* (YFP fusion at N-terminus of *Rpb1* gene) constructs containing different number of K7-to-S7 mutations (*Figure 2a*). Mutant 0K does not have any K7 residues and therefore resembles a yeast-like CTD, but with 52 repeats. Mutant 1K retains only one K7, on repeat 35, which is conserved in *D. melanogaster* (aligning from C-terminus of CTD) and is the only K7 residue present in *C. elegans*. Mutant construct 3K has three K7 residues present at repeats 35, 40 and 47, all of which are conserved in *D. melanogaster*. Finally, we used a wild-type murine *Rpb1* construct (8K), which contains all eight vertebrate-conserved K7 residues, as a control for expression and α-amanitin selection. We produced viable mouse NIH-3T3 fibroblast lines that express each of the four constructs and show stable YFP-RPB1 expression for more than one month in culture and after several passages under α-amanitin selection. Viability of cells expressing α-amanitin-resistant RPB1 was previously shown for K7-to-R7 mutations (*Schroeder et al., 2013*) or for other CTD constructs without all lysines, where *Rpb1* contained only consensus heptapeptide repeats (*Chapman et al., 2005*; *Hintermair et al., 2012*).

## Mutation of CTD-K7 residues is compatible with CTD phosphorylation

To determine whether non-canonical CTD-K7 residues are important for CTD phosphorylation, we performed western blotting using total protein extracts from stable NIH-3T3 clones expressing 8K (wild-type) or 0K constructs (*Figure 2b*, *Figure 2—figure supplement 1*); extracts from untransfected NIH-3T3 fibroblasts were analyzed as an additional control. Total expression levels of YFP-RPB1 fusion proteins, detected using an antibody against the N-terminus of RPB1, were similar to the levels of endogenous RPB1 in the parental NIH-3T3 cell line. As expected, YFP-RPB1 fusion proteins migrate at a higher molecular weight than endogenous RPB1, confirmed using antibodies that detect the YFP tag (*Figure 2b*, *Figure 2—figure supplement 1*). Western blot analyses of CTD phosphorylation using highly specific antibodies against S5p, S7p and S2p (*Brookes et al., 2012*; *Stock et al., 2007*), detect hyperphosphorylated (II0) RPB1 in untransfected NIH-3T3, wild-type 8K and 0K mutant cells, showing that mutation of K7-to-S7 residues is compatible with normal global levels of serine phosphorylation.

To examine the effect of K7-to-S7 mutation on the subcellular localization of RPB1, we used confocal microscopy and YFP fluorescence to detect YFP-RPB1 fusion proteins and found that the typical RNAPII nucleoplasmic distribution is unaffected by K7-to-S7 mutations (*Figure 2c*; e.g. *Xie et al., 2006*). Immunofluorescence using S5p antibodies also shows similar distribution and levels of S5p in 8K and 0K cells (*Figure 2c*). These observations show that mutation of CTD-K7 residues is compatible with viability of mouse fibroblasts, and suggest that global serine phosphorylation and RNAPII localization are independent of the presence or absence of K7 residues.

## CTD-K7 residues are methylated in vivo

Acetylation of CTD-K7 residues was recently identified and found associated with inducible gene expression (*Schroeder et al., 2013*). Lysine acetylation has been extensively studied in the context of histone proteins, where it is often counter-balanced by methylation, with clear roles in regulation of gene expression and repression (*Bannister and Kouzarides, 2011*; *Wozniak and Strahl, 2014*). To investigate whether K7 methylation could counteract K7 acetylation of RNAPII, we developed specific monoclonal antibodies using CTD peptides methylated on K7. With the aim of raising antibodies that could potentially detect methylation in several or all of the K7-containing CTD repeats, we chose the peptide sequence centered on the K7 residue in repeat 35 of the CTD (*Figure 3a*).

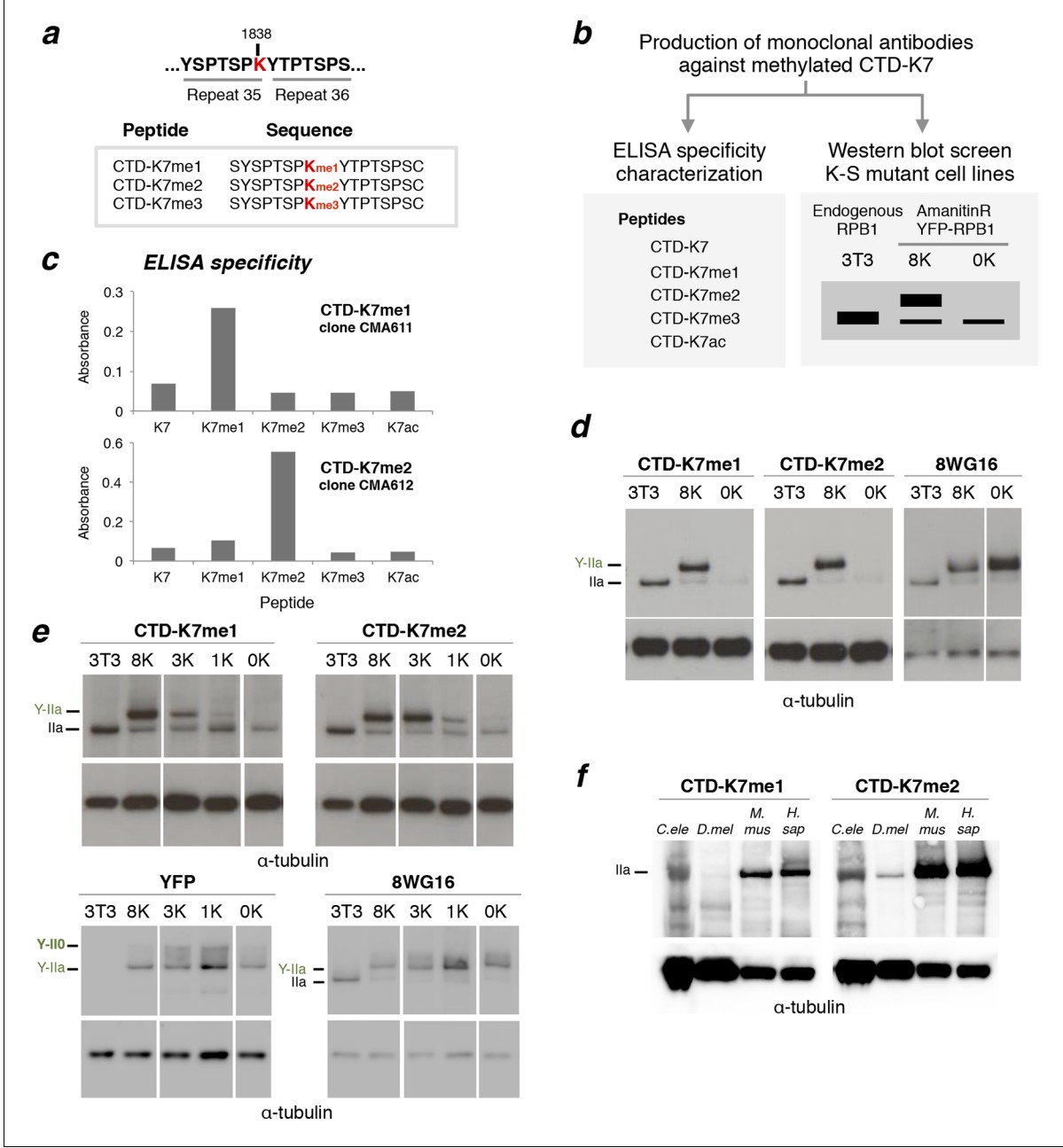

**Figure 3.** RPB1 is mono- and di-methylated at CTD-K7 residues. (**a**) Amino-acid sequence of CTD-K7-methyl peptides used for immunization, designed based on the sequence of mouse CTD repeats 35 and 36. (**b**) Schematic representation of strategy used for production and screening of specific CTD-K7-methyl antibodies. Antibody clones that specifically recognize K7 or its modifications should bind strongly to the wild-type band, the 8K slower-migrating band, but not to the mutant 0K band. (**c**) Specificity of CTD-K7 methyl antibodies was assessed by ELISA using unmodified (K7), mono- (K7me1), di- (K7me2), tri-methylated (K7me3) and acetylated (K7ac) CTD peptides (*Table 1*). Clones CMA611 and CMA612 are specific for K7me1 and K7me2, respectively. (**d**) K7me1 and K7me2 mark hypophosphorylated RPB1 in mouse cells with migration similar to forms detected using 8WG16 antibody. Western blotting was performed using total protein extracts from NIH-3T3 (3T3), and from NIH-3T3 cells stably expressing wild-type 8K (8K) or mutant 0K (0K). K7me1 and K7me2 are detected in 3T3 and 8K, but not in 0K cell lines. CTD methylation migrates at the level of hypophosphorylated RNAPII (IIa and Y-IIa). Low levels of methylation of endogenous RPB1 is also detected in 8K and 0K cell lines, due to expression from endogenous Rpb1 locus. α-Tubulin was used as loading control. For 8WG16 blot, samples were run in the same gel, and re-ordered to improve clarity. Original western blots are shown in *Figure 3—figure supplement 1a*. (**e**) CTD K7 residues are mono- and di-methylated in 3T3 cells at levels that increase with K7 number. K7me1 and K7me2 were detected by western blotting using whole-cell extracts from 3T3 lines expressing 8K, 3K, 1K or 0K Rpb1 constructs; untransfected 3T3 cell extracts were used as an additional control. RPB1 levels were measured by immunoblotting of YFP and using 8WG16 antibody with specificity for unmodified S2. α-Tubulin was used as loading control. Samples were run in the same gel, and re-ordered to improve clarity. Original

*Figure 3 continued on next page*

*Figure 3 continued*

western blots are shown in *Figure 3—figure supplement 1c*. (f) K7me2 and K7me1 are detected in invertebrates, mouse and human cells. Western blotting of K7me1 and K7me2 was performed using *C. elegans* whole worm extract (*C. ele*), *D. melanogaster* embryo extract (*D. mel*), total cell extracts from NIH-3T3 cells (*M. mus*) and from human HEK-293T cells (*H. sap*). α-Tubulin was used as loading control. RPB1, RNA polymerase II large subunit; YFP, yellow flourescent protein. Original western blots are shown in *Figure 3—figure supplement 1d*.

The following figure supplement is available for figure 3:

**Figure supplement 1.** Complete western blots used in *Figure 3a, 3e and 3f*, and detection of CTD methylation in mouse ES cells.

This heptad has the most represented K7-repeat sequence (YSPTSPK), and is the K7 residue with most conserved distance to the C-terminal end of RPB1 in vertebrates and invertebrates (*Figure 1b*). Peptides modified by mono-, di- and tri-methylation of K7 residues were used for immunization. The supernatants of hybridoma clones were screened by enzyme-linked immunosorbent assay (ELISA) to test for specificity to CTD-K7 methylation (*Figure 3b*). We identified several antibody clones specific for CTD-K7 mono- or di-methylation (*Figure 3c*). Clone CMA611 is specific for CTD-K7me1 and does not recognize unmodified CTD-K7, CTD-K7me2, CTD-K7me3 or CTD-K7ac. Clone CMA612 is specific for CTD-K7me2 and does not bind to the other peptides tested (*Figure 3c*). Although ELISA analyses identified clones that recognize CTD-K7me3, or both CTD-K7me3 and me1/2 forms, these clones showed reactivity towards other proteins (not shown). We therefore did not perform further analyses using CTD-K7me3 antibody clones.

To test whether mono- or di-methylation of the CTD could be identified in vivo, we performed western blotting on total extracts from NIH-3T3, 8K and 0K cell lines using the novel antibodies against CTD-K7me1 and CTD-K7me2 (*Figure 3d*, *Figure 3—figure supplement 1a*). K7me1 and K7me2 are detected in NIH-3T3 and in 8K cell lines, both of which express a CTD with eight K7 residues, but not in the 0K mutant cell line, where all K7 residues are mutated to S7. These results confirm the specificity of the K7me1 and K7me2 antibodies to CTD-K7 modifications. The single band in NIH-3T3 and in 8K cell lines shows lack of cross-reactivity to other NIH-3T3 proteins (*Figure 3—figure supplement 1a*). These results demonstrate the existence of *in vivo* methylation of non-canonical K7 residues of the CTD in mouse fibroblasts.

RPB1 migrates in two major forms, a fast migrating hypophosphorylated (IIa) state and a slower migrating hyperphosphorylated (II0) state, as well as intermediate phosphorylation forms. Interestingly, we found that both the K7me1 and K7me2 antibodies detect the hypophosphorylated RPB1 and YFP-RPB1 bands (*Figure 3d*). This band is also detected by the antibody 8WG16, which preferentially recognizes unmodified S2 residues (reviewed in*Brookes and Pombo, 2009*). To confirm the presence of K7me1 and K7me2 within hypophosphorylated RPB1, we repeated the K7me1 and K7me2 western blotting in mouse ES cells, confirming immunoreactivity to hypophosphorylated RPB1 (*Figure 3—figure supplement 1b*).

## Multiple CTD-K7 residues are methylated

To explore the extent of methylation of the eight mammalian K7 residues, we performed western blots using the NIH-3T3 cell lines engineered to express YFP-RPB1 fusion proteins bearing different numbers of K7 residues (0, 1, 3 or 8 lysines; *Figure 3e*, *Figure 3—figure supplement 1c*). Mono- and di-methylation were identified in total cell extracts from the 1K cell line. The intensities of mono- and di-methylation increase in the 3K line, indicating that several lysine residues are mono- and di-methylated in the same CTD. The level of mono-methylation increases further in the 8K-cell line, showing abundant mono-methylation of the CTD in vivo. In contrast, the di-methylation levels remain similar between the 8K and 3K lines, suggesting that not all eight CTD lysine-7 residues are simultaneously di-methylated, and reflecting a possible preference for di-methylation of the K residues conserved between mammals and invertebrates, which are present in the 3K construct. Similar expression levels of YFP-RPB1 were confirmed in the four cell lines using western blots for YFP and 8WG16 (*Figure 3e*).

## CTD-K7 residues are also methylated in human cells, *D. melanogaster* and *C. elegans*

We next tested whether CTD-K7 methylation is conserved across species. K7me2 is also detected in whole protein extracts from adult *Caenorhabditis elegans* worm, *Drosophila melanogaster* embryos, and human HEK293 cells, in western blotting analysis (*Figure 3f*, *Figure 3—figure supplement 1d*). These observations reveal, for the first time, conservation of a non-consensus CTD modification between vertebrates and invertebrates. K7me1 also occurs in human cells and *C. elegans*, but is not easily detected in extracts from *D. melanogaster* embryos, suggesting that di-methylation may be a more prevalent methylation mark. Detection of mono- and di-methylation at the single *C. elegans* K7 residue, with antibodies produced using peptides based on the mammalian repeats 34-–35, also suggests that the recognition of K7 methylation is in general robust to small differences in the amino-acid sequences that flank the modified K7 residues (compare *C. elegans* sequence, -$\underline{S_4}$-$S_5$-$P_6$-$K_7$-$Y_1$-$S_2$-$P_3$-, with immunizing mammalian peptide sequence, -$T_4$-$S_5$-$P_6$-$K_7$-$Y_1$-$\underline{T_2}$-$P_3$-; *Figure 1a, b*). This conclusion is also supported by the increased detection of K7me1 and K7me2 with increased number of K7 residues in the mammalian CTD (cell lines 1K, 3K and 8K; *Figure 3e*), each flanked by slightly different amino acids (*Figure 1a*).

## CTD-K7 methylation occurs early during the transcriptional cycle

The observation that mono- and di-methylation of CTD-K7 residues is detected primarily in the hypophosphorylated (faster-migrating) forms of RPB1 (*Figure 3d*) suggests that CTD methylation is associated with early stages of the transcription cycle. However, it could also result from steric hindrance of K7me1 and K7me2 antibody binding by CTD phosphorylation. To test whether CTD phosphorylation interferes with immunodetection of K7 methylation, we performed western blots from total protein extracts obtained from mouse ES cells, and pre-treated the blots with alkaline phosphatase to remove phosphoepitopes prior to immunoblotting (*Figure 4a*, *Figure 4—figure supplement 1a*). We find that the detection of K7 methylation remains specific to the hypophosphorylated RPB1 after treatment of immunoblots with alkaline phosphatase, showing only a minor increase in the detection of K7me1 and K7me2 at intermediately phosphorylated forms, in conditions that fully abrogate detection of phosphorylated epitopes (e.g. S5p; see also *Stock et al., 2007*). Therefore, immunodetection of K7me1 and K7me2 is only minimally affected by CTD phosphorylation, suggesting that K7me1 and K7me2 modifications are depleted from elongation-competent hyperphosphorylated RPB1 complexes. Interestingly, the association of K7me1 and K7me2 with hypophosphorylated form of RPB1 differs from K7ac, previously shown to occur at both hypo- and hyperphosphorylated RPB1 forms (*Schroeder et al., 2013*), suggesting that K7 methylation may precede K7 acetylation during the transcription cycle.

To further investigate whether K7 mono- and di-methylation occur upstream of elongation, we treated ES cells with flavopiridol, an inhibitor of RNAPII elongation (*Chao and Price, 2001*). Depletion of elongation-competent complexes can be achieved by short treatment of ES cells with flavopiridol (10 μM, 1 hr), as shown by loss of S2p detection and lower mobility of S5p forms in western blots (*Stock et al., 2007*; *Figure 4b*, *Figure 4—figure supplement 1b*). We find that K7me1 and K7me2 levels are only minimally increased by flavopiridol treatment (*Figure 4b*), consistent with both modifications being associated with pre-elongation stages of transcription. The minor increase in K7me1 and K7me2 levels agrees with the slightly increased detection of K7 methylation upon CTD dephosphorylation.

We then tested whether K7me1 and K7me2 are localized within the nucleus using immunofluorescence in mouse NIH-3T3 cells (*Figure 4c*). We find K7me1 and K7me2 concentrated in punctate nucleoplasmic domains, absent from nucleoli and regions of heterochromatin. The K7me1 and K7me2 foci are sparser than the discrete nucleoplasmic domains containing total YFP-RPB1 or S5p (see *Figure 2c*), consistent with their presence at RNAPII complexes involved in more restricted transcription events, rather than extensively marking chromatin-free RPB1.

## CTD-K7 mono- and di-methylation mark promoters of active genes

To explore the role of K7me1 and K7me2 in the transcription cycle, we mapped their chromatin occupancy genome-wide using chromatin immunoprecipitation coupled to next generation sequencing (ChIP-seq) in mouse ES cells (*Figure 5*). The chromatin occupancy of K7me1 and K7me2

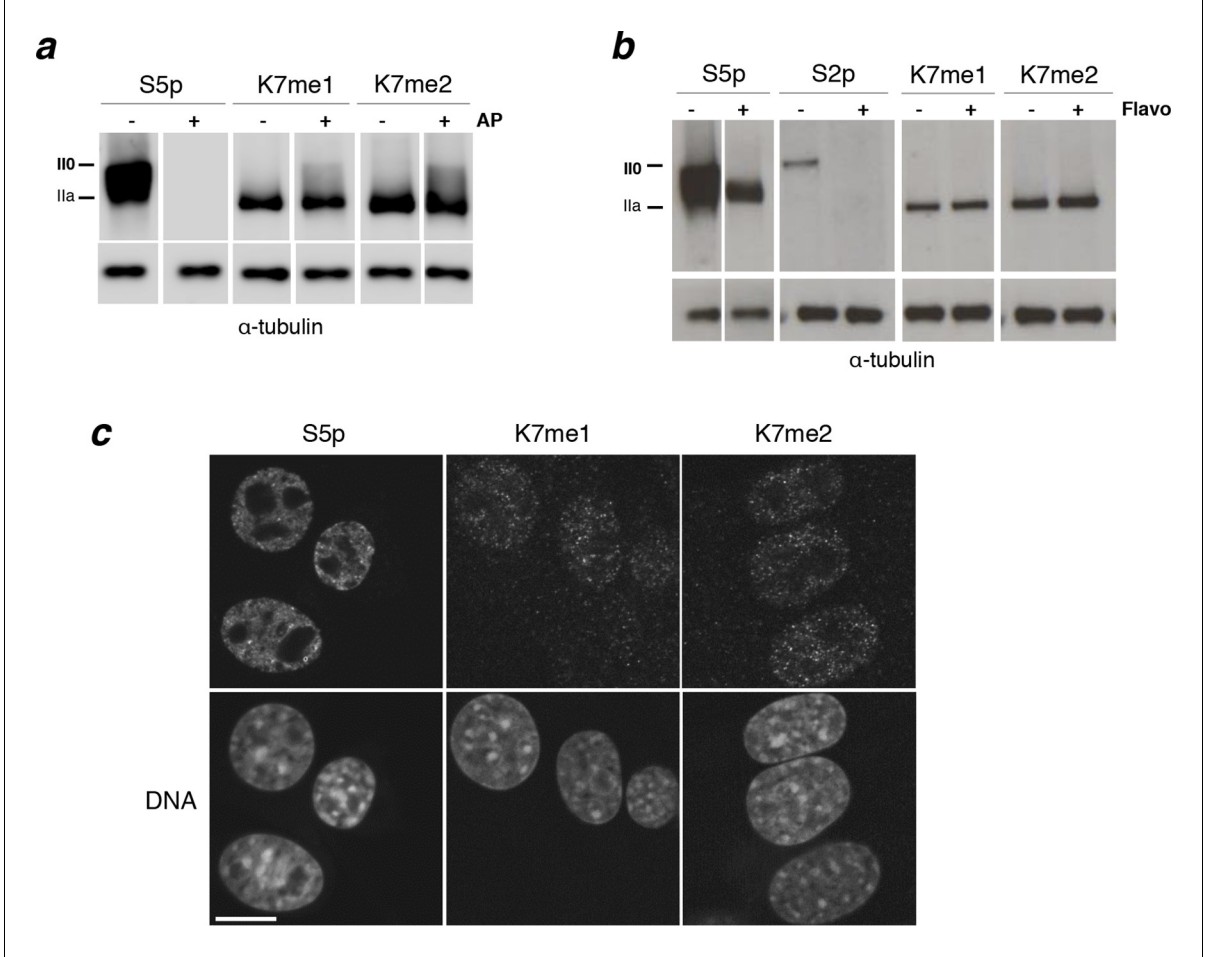

**Figure 4.** Interplay between K7me1 and K7me2 with RPB1 phosphorylation. (**a**) CTD K7me1 and K7me2 mark hypophosphorylated and intermediately phosphorylated Rpb1 forms, but not the hyperphosphorylated (II0) form. Western blotting using the indicated antibodies was performed after treatment of nitrocellulose membranes in the presence (+) or absence (–) of alkaline phosphatase (AP). Hypo- (IIa) and hyperphosphorylated (II0) RPB1 forms are indicated. α-Tubulin was used as loading control. Lanes were re-ordered to improve clarity. Original western blots are shown in *Figure 4— figure supplement 1a*. (**b**) K7me1 and K7me2 abundance is insensitive to CDK9 inhibition with inhibitor flavopiridol. Mouse ES cells were treated with flavopiridol (10 μM, 1 hr), before western blotting using antibodies specific for S5p, S2p, K7me1 or K7me2. Hypo- (IIa) and hyperphosphorylated (II0) RPB1 forms are indicated. α-Tubulin was used as loading control. Lanes were re-ordered to improve clarity. Original western blots are shown in *Figure 4—figure supplement 1b*. (**c**) K7me1 and K7me2 are localized in the nucleoplasm with a more restricted distribution than S5p. Whole-cell immunofluorescence of S5p, K7me1 and K7me2 was performed using mouse NIH-3T3 fibroblasts. Nucleic acids were counterstained with TOTO-3. Scale bar, 10 μm. RPB1, RNA polymerase II large subunit.

The following figure supplement is available for figure 4:

**Figure supplement 1.** Complete western blots used in *Figure 4a and 4b*.

was compared with RNAPII phosphorylation (S5p, S7p, S2p and unmodified S2 detected with antibody 8WG16), with K7 acetylation, and with mRNA-seq, using published datasets from mouse ES cells (*Figure 5a*; *Brookes et al., 2012*; *Schroeder et al., 2013*). S5p, S7p and 8WG16 are primarily enriched at gene promoters and downstream of polyadenylation sites; S2p is detected along gene bodies and is most highly enriched immediately after polyadenylation sites (*Brookes et al., 2012*). CTD-K7ac occupies gene promoters and extends into gene bodies, as previously described (*Schroeder et al., 2013*).

Inspection of ChIP-seq profiles at single genes shows that K7me1 and K7me2 are highly enriched at promoters of active genes (*Eed, Rpl13*, and *Tuba1a*), and are detected at lower levels beyond transcription end sites (*Figure 5a*). K7ac is also found at gene promoters but is often present

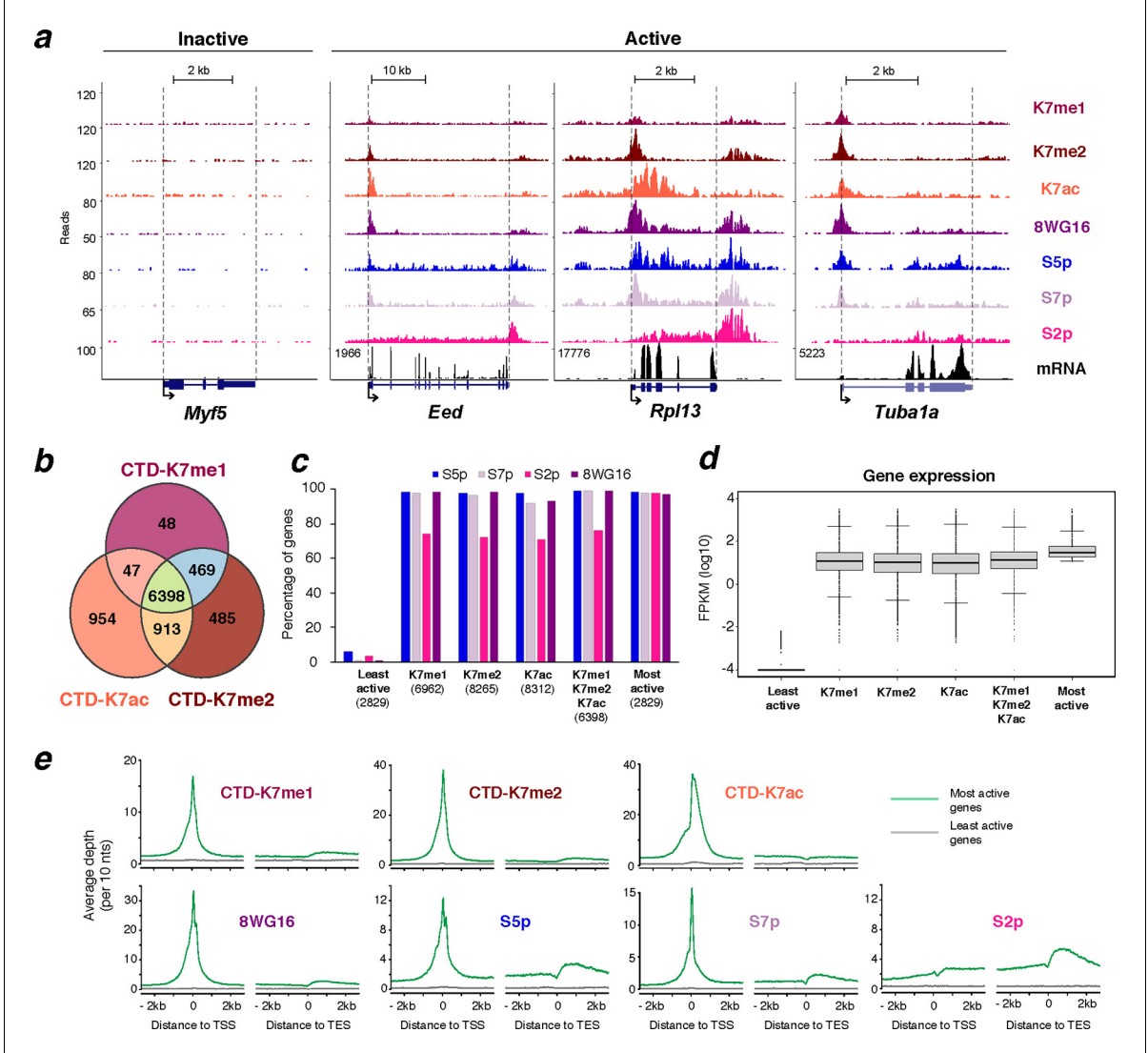

**Figure 5.** K7me1 and K7me2 mark promoters of expressed genes. (a) K7me1 and K7me2 are enriched at promoters of active genes. ChIP-seq profiles for K7me1, K7me2, K7ac, 8WG16, S5p, S7p and S2p, and mRNA-seq profiles are represented for the inactive gene *Myf5*, and active genes *Eed, Rpll13* and *Tuba1a*. Images were obtained from UCSC Genome Browser using mean as windowing function. (b) Methylation and acetylation of CTD-K7 residues coincides at most genes. Gene promoters positive for K7me1 (6962), K7me2 (8265) and K7ac (8312) were identified using a peak finder approach (see Materials and methods). The overlap between the three CTD-K7 modifications is represented using a Venn diagram. (c) CTD-K7 methylation and acetylation extensively co-occur with other CTD modifications. The percentages of genes positive for S5p, S7p, S2p and unmodified S2 are represented for each group of genes positive for K7me1, K7me2 and K7ac individually or simultaneously. Least and most active genes (bottom and top 15% expressed genes, respectively) are represented for comparison. Numbers of genes are indicated below each group of genes. (d) CTD-K7 methylation and acetylation are associated with active genes genome-wide. mRNA levels are similar at genes positive for K7me1, K7me2 and/or K7ac. Least and most active genes are represented for comparison. A pseudocount of $10^{-4}$ was added to FPKM prior to logarithmic transformation. (e) K7me1, K7me2 and K7ac are strongly enriched around the TSS of most active genes. ChIP-seq average enrichment profiles of K7me1, K7me2, K7ac, unmodified S2 (8WG16), S5p, S7p and S2p, for the most active (green) and least active (gray) genes represented over 5 kb window centered on the TSS and TES. Genes positive for H3K27me3 and/or H2Aub1 were excluded, to minimize confounding effects due to polycomb repression, which is abundant in mouse ES cells (*Brookes et al., 2012*). CTD, C-terminal domain; ES, embryonic stem; FPKM, Fragments Per Kilobase of exon per Million mapped reads; mRNA, messenger RNA; TES, transcription end sites; TSS, transcription start sites.

The following figure supplements are available for figure 5:

**Figure supplement 1.** CTD-K7 mono- and di-methylation are enriched at promoters of active genes.

**Figure supplement 2.** RNAPII CTD is mono- and di-methylated exclusively at active genes and not at Polycomb repressed genes.

downstream of K7 methylation occupancy, within gene bodies. K7 methylation and acetylation, and other RNAPII modifications, are not detected at inactive genes (e.g. *Myf5* gene). K7me1 and K7me2 occupancy at active genes, and absence at inactive, were confirmed by single gene quantitative polymerase chain reaction (qPCR) (*Figure 5—figure supplement 1a*). In summary, we find that K7me1 and K7me2 are more tightly localized to gene promoters than K7ac, suggesting association with the earliest stages of transcription.

Given that K7me1, K7me2 and K7ac exhibit different occupancy patterns at active genes, we assessed the genome-wide promoter distribution and abundance of each mark. We first identified all genomic regions positive for CTD-K7 marks, using a peak finder suited for both sharp and broader occupancy patterns (Bayesian change-point, BCP; *Xing et al., 2012*). We then used these positive regions to classify gene promoters according to the presence of each modification (*Figure 5—figure supplement 1b*). A total of 6962, 8265 and 8312 gene promoters were classified as positive for K7me1, K7me2 and K7ac, respectively. The vast majority of these genes are (i) positive for all three marks (n = 6398; *Figure 5b*), (ii) also marked by S5p, S7p, S2p and 8WG16 (*Figure 5c*), and (iii) transcriptionally active at the messenger RNA (mRNA) level (*Figure 5d*). As a further test of the specificity of K7me1 and K7me2 presence at productive RNAPII complexes, we evaluated their presence at polycomb repressed (PRCr) genes. This group of genes is associated with poised RNAPII complexes in mouse ES cells, which are characterized by an unusual CTD state phosphorylated at S5 but not at S7 or S2, and that does not lead to mRNA expression (*Brookes et al., 2012*; *Stock et al., 2007*; *Tee et al., 2014*). Consistent with the specificity of K7me1 and K7me2 to expressed genes, we found that K7me1 and K7me2 do not mark poised RNAPII at PRCr genes marked by H3K27me3 and H2Aub1 (*Figure 5—figure supplement 2a and 2b*) and that they are associated with promoters marked by H3K4me3 (*Figure 5—figure supplement 2c*).

To ascertain the distribution of the different CTD modifications at active genes, we plotted average occupancy profiles around transcription start sites (TSS) and transcription end sites (TES) of the 15% most and least expressed genes (*Figure 5e*). The novel CTD modifications K7me1 and K7me2 are sharply enriched at the TSS. These profiles are very similar to that of S7p while, in contrast, K7ac is more broadly localized up- and downstream of the TSS. Together with the observations that K7me1 and K7me2 mark hypophosphorylated RPB1 and that their abundance is insensitive to inhibition of RNAPII elongation (*Figure 4a,b*), their specific enrichment at the promoters of active genes suggests that they mark the earliest stages of the transcription cycle at protein coding genes.

## K7me1 and K7me2 at gene promoters have negative contributions to mRNA levels

To explore the temporal sequence of CTD modifications during the transcription cycle and how they predict gene expression levels, we performed extensive correlation analyses between different RNA-PII modifications (*Figure 6*). We focused on active genes using a refined group of genes which are (i) positive for S5p, S7p and S2p, (ii) expressed at the mRNA level, and (iii) negative for the repressive histone modifications H3K27me3 and H2Aub1. We also excluded overlapping genes and genes where the maximum peak of RNAPII (marked by unphosphorylated RNAPII using 8WG16 antibody) is not within 50 bp of the annotated TSS, yielding a group of 1564 active genes. We find that mRNA levels are most strongly correlated with S2p levels at TES of active genes (Spearman's correlation coefficient 0.62), followed by promoter K7ac (0.55) and S7p (0.49; *Figure 6a*). In contrast, the enrichment levels of K7me1 and K7me2 at gene promoters (which are highly self-correlated, 0.93) poorly correlate with S2p or mRNA (0.31–0.45), but highly correlate with modifications associated with initiation and early elongation (unphosphorylated RNAPII, S5p and S7p; 0.72–0.86). These correlations suggest that CTD-K7 methylation is related to RNAPII occupancy at gene promoters, and not directly to elongating complexes. Similar results are obtained with a more inclusive correlation analysis using the full list of non-overlapping active genes irrespective of 8WG16 peak distance to TSS (n = 4271), and including histone modifications and CpG content (*Figure 6—figure supplement 1a*).

Given that all CTD modifications correlate with each other to a certain extent, we performed partial correlation analyses to disentangle the association between K7 methylation and productive elongation after accounting for the effects of other intervening RPB1 modifications (*Figure 6b*). Surprisingly, we found that the partial correlation between K7me2 and S2p becomes zero after removing individual contributions of S5p or K7ac; similar results were obtained for K7me1 (*Figure 6—figure supplement 1b*). Moreover, removing the contribution of S7p alone, or the combined

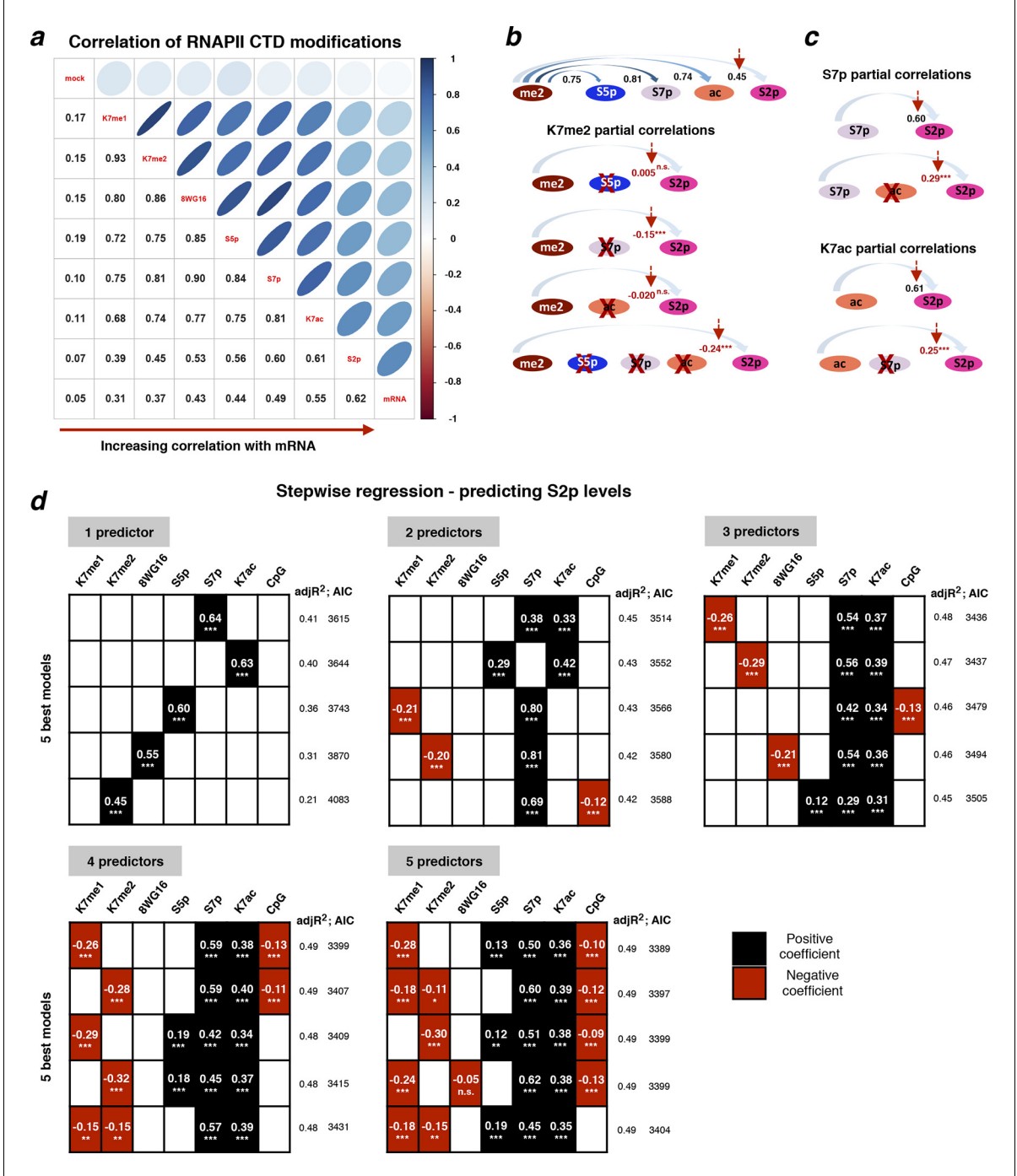

**Figure 6.** Exploring the relationship between different CTD modifications using correlation and linear regression analyses. (a) Matrix of Spearman's correlation coefficients between the levels of K7me1, K7me2, K7ac, 8WG16, S5p, S7p, S2p (2 kb window after TES), mRNA and mock ChIP control ordered according to increasing correlation with mRNA. This correlation analysis was performed with the group of active genes (n = 1564) positive for S5p, S7p and S2p, with expression level of FPKM >1, and negative for H3K27me3 and H2Aub1, also excluding overlapping genes and genes whose maximum RNAPII peak (8WG16) is >50 bp away from the annotated TSS. *Figure 6—figure supplement 1* represents the full group of active genes, including genes with maximum 8WG16 peak deviated from TSS >50 bp. (b) The partial correlations between K7me2 and S2p is zero or becomes negative after removing the contribution of other intervening modifications. Schematic summarizes the dependencies between K7me2 and other CTD modifications relative to S2p (TES); the respective correlations are represented on top. Partial correlations between K7me2 and S2p after removing the effect of the other CTD modifications are indicated in red (*** $P$-value < $1\times10^{-9}$; n.s., non-significant). (c) Levels of S7p and K7ac independently contribute to S2 phosphorylation. The partial correlations between S7p and S2p when controlling for K7ac (top) and between K7ac and S2p when controlling for S7p (bottom) remain positive (indicated in red; *** $P$-value < $1\times10^{-9}$). (d) Exhaustive stepwise regression analysis for prediction of S2p

*Figure 6 continued on next page*

*Figure 6 continued*

levels at the TES using K7me1, K7me2, 8WG16, S5p, S7p, K7ac and CpG. The five best models using 1– 5 predictors are shown. Positive and negative coefficients are represented by black and red squares, respectively; the values of adjusted $R^2$ and Akaike information criterion (AIC) are indicated for each model (* *P*-value $\leq$ 0.05; ** *P*-value $\leq$ 0.01; *** *P*-value $\leq$ 0.001; n.s., non-significant). ChIP, chromatin immunoprecipitation; CTD, C-terminal domain; FPKM, Fragments Per Kilobase of exon per Million mapped reads; mRNA, messenger RNA; TES, transcript end sites; TSS, transcription start sites.

The following figure supplements are available for figure 6:

**Figure supplement 1.** Correlations between different RNAPII CTD modifications and histone marks for all active genes.

**Figure supplement 2.** CTD K7me1 and K7me2 negatively contribute for the prediction of S2p levels.

contributions of S7p, S5p and K7ac, results in small negative correlations of K7me2 (and K7me1) with S2p (*Figure 6—figure supplement 1b*). These results suggest that K7me1 and K7me2 are anti-correlated with S2p, which in turn is highly correlated with K7ac. We also performed partial correlation analyses to test the individual contributions of S7p and K7ac to productive elongation after adjusting for the effects of their correlation with each other. Both marks maintain positive partial correlations with S2p levels (0.29 and 0.25 respectively; *Figure 6c*), suggesting that both S7p and K7ac have direct links to the extent of elongation.

In these pairwise correlation analyses, we began to dissect unexpected dependencies between different CTD modifications and how they relate to productive elongation. However, the most likely scenario is that several CTD modifications jointly contribute to expression levels. To further explore whether CTD modifications work together to promote productive transcription, we employed stepwise regression models to determine the CTD modifications that best predict elongation marked by S2p levels, for a given number of predictors (*Figure 6d*). These analyses confirm that S7p and K7ac are (i) the best predictors of S2p in single-variable models, (ii) the top combination of modifications in two-variable models, and (iii) always present in the top models when considering three, four and five modifications. Surprisingly, we also find that the promoter levels of K7me1 and K7me2 have significant negative contributions to the S2p elongation mark and mRNA expression, when considered together with the positively correlated variables S5p, S7p and/or K7ac. The dependencies identified were further tested using LASSO regression, which is more robust to artifacts (due to co-linearity in the predictors and over-fitting), in combination with a common cross-validation approach (minimum error plus one standard error criterion) to select the most parsimonious linear model in the LASSO path (*Figure 6—figure supplement 2*). We confirmed major positive contributions of S7p and K7ac to the levels of S2p (and mRNA), together with smaller (non-redundant) positive contribution of S5p, and negative contribution of K7me1. LASSO regression also shows that the levels of K7me1 and K7me2 are redundant, as K7me2 has a similar negative contribution to the minimal linear model found by LASSO analysis, when K7me1 is excluded from the variable set (not shown). Taken together, these results reinforce the observation that K7me1 and K7me2 are early transcription marks that are present at actively transcribed genes, but have negative contribution to mRNA levels, and raise the possibility that a balance between K7 methylation and K7 acetylation might define the extent of productive elongation at active genes.

## The balance between CTD-K7 methylation and acetylation correlates with gene expression

To investigate whether the balance between CTD-K7 methylation and acetylation at gene promoters could be predictive of gene expression, we calculated the ratio between K7me2 and K7ac and investigated how it relates to mature and nascent RNA levels. We find that the K7me2/K7ac ratio negatively correlates with mRNA expression (Spearman's correlation coefficient -0.35; *Figure 7a*, *Figure 7—figure supplement 1a*). Analyses of nascent transcription using published GRO-seq (global run-on sequencing) data (*Jonkers et al., 2014*) yield a similar negative correlation with coefficient –0.31 (*Figure 7—figure supplement 1b*).

To visualize the effect of the ratio between K7 methylation and acetylation on promoter occupancy and expression of single genes, we generated ChIP-seq heatmaps centered on active

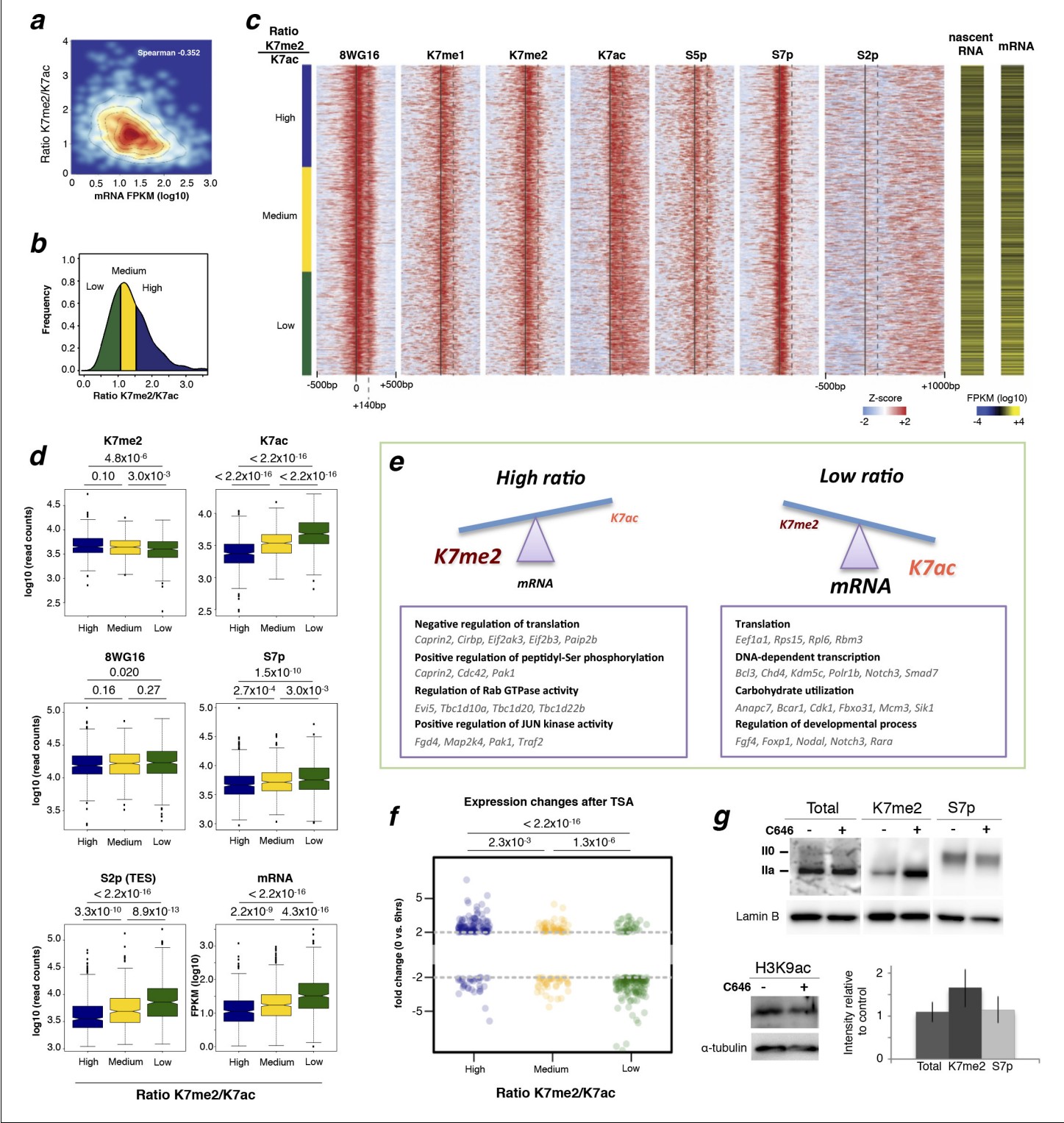

**Figure 7.** CTD-K7 methylation and acetylation have different distributions at the promoters of active genes and their levels are associated with gene expression. (a) K7me2/K7ac ratio correlates negatively with mRNA expression. (b) Distribution of K7me2/K7ac ratios, and their division into three quantiles: low (green), medium (yellow) and high (blue) ratios. (c) Heatmaps showing ChIP-seq density for a group of active genes (n = 1564 genes, defined as in *Figure 6a*) using ± 500 bp windows centered on promoters, except for S2p, using window -500 to +1000 bp; z-scores per gene are represented. Genes were ordered according to K7me2/K7ac ratios, from highest to lowest. Gene expression (both nascent and mature RNA; FPKM) is represented for comparison. (d) Unmodified S2 (8WG16), K7me2, K7ac, S7p, S2p (TES) and gene expression (mRNA) levels are represented for the

*Figure 7 continued on next page*

*Figure 7 continued*

three quantiles of K7me2/K7ac ratio. A Wilcoxon rank-sum test was used to calculate significant differences between the K7me2/K7ac quantiles and the respective p-values are represented. (e) Schematic representation of the relation between K7me2 and K7ac levels with mRNA expression. Examples of Gene Ontology (GO) terms and respective genes associated with high and low K7me2/K7ac ratios are represented. (f) Amount of fold change after 6 hr treatment with the histone deacetylase inhibitor TSA is represented for the three quantiles of K7me2/K7ac ratio. Fisher's exact test was used to calculate significant differences between the K7me2/K7ac quantiles and the respective P-values are represented. Only genes with a minimum fold change of 2-fold change at 6 hr TSA treatment are shown. Dashed line: 2-fold change. (g) Inhibition of P300 promotes an increase in global levels of CTD-K7me2. Mouse ES cells were treated with P300 inhibitor C646 (30 µM, 3 hr), before western blotting with antibodies specific for total RNAPII, K7me2 and S7p (top panel) and H3K9 acetylation (bottom left panel). Hypo- (IIa) and hyperphosphorylated (II0) RPB1 forms are indicated. Lamin B and α-tubulin were used as loading control. Original western blots are shown in *Figure 7—figure supplement 4*. Relative quantification of western blot signal intensity (bottom right panel). Signal intensity of total RNAPII, K7me2 and S7p for control and C646 treated samples was normalized to the corresponding Lamin B signal. The levels of total RNAPII, K7me2 and S7p after P300 inhibition are represented relative to the signal from control (DMSO only) cells. Bars represent average values with standard deviation from two biological replicates and two to four technical replicates. CTD, C-terminal domain; ChIP-seq, chromatin immunoprecipitation with sequencing; DMSO, dimethyl sulfoxide; ES, embryonic stem; FPKM, Fragments Per Kilobase of exon per Million mapped reads; mRNA, messenger RNA; TSA, Trichostatin A.

The following figure supplements are available for figure 7:

**Figure supplement 1.** CTD methylation / acetylation ratio negatively correlates with S2p and RNA production.

**Figure supplement 2.** Distribution of different RPB1 modifications around the promoter of active genes.

**Figure supplement 3.** Genes with higher K7me2/K7ac ratio are up-regulated after TSA treatment.

**Figure supplement 4.** Complete western blots used in *Figure 7g*.

---

promoters after ordering genes according to their K7me2/K7ac ratio and dividing them in three groups (low, medium and high ratio; *Figure 7b*). We observe that unmodified S2, marked by 8WG16, shows a major peak centered on the TSS, as well as a secondary peak 140 bp downstream, and is independent of the K7me2/K7ac ratio (*Figure 7c*; see also *Figure 7—figure supplement 2a* for the full list of non-overlapping active genes). K7me1 and K7me2 are enriched at the TSS-centered peak, but not at the secondary 8WG16 peak +140 bp downstream of the TSS (*Figure 7c*), which has previously been associated with promoter-proximal pausing (*Quinodoz et al., 2014*). In contrast, K7ac expands downstream of the TSS up to +500 bp, especially at genes with low K7me2/K7ac ratios, but without specific enrichment at the secondary 8WG16 peak. S2p, which is a broad elongation mark enriched mostly at the TES, does not peak at the TSS. Interestingly, genes with the highest K7me2/K7ac ratios exhibit the lowest mRNA or nascent transcript expression (*Figure 7c and 7d*), and lowest S2p in the beginning of gene bodies (*Figure 7c*) and after TES (*Figure 7d*). Inspection of single gene profiles confirms these general observations (*Figure 7—figure supplement 2b*).

To explore the functionality of genes associated with higher or lower K7me/K7ac ratios, we performed Gene Ontology (GO) analyses (*Figure 7e*). Interestingly, we found that genes with high K7me2/K7ac ratios are associated with GO terms that relate with *negative regulation of translation*, and kinase cascades, such as *positive regulation of peptidyl-serine phosphorylation*, of *Rab GTPase activity* and of *JUN kinase activity*. In contrast, genes with low K7me2/K7ac ratios are associated with GO terms that relate with housekeeping functions in mouse ES cells such as *translation, DNA-dependent transcription, carbohydrate utilization* and *regulation of developmental process*. These results suggest that the balance between K7 methylation and acetylation at gene promoters is important to define the extent of productive transcription at groups of genes with different biological functions, namely in the transition between initiation and elongation, which can contribute to fine-tuning gene expression levels.

Previous work reported transcriptional up-regulation of gene expression upon treatment of ES cells with histone deacetylase inhibitor Trichostatin A (TSA; *Karantzali et al., 2008*), which has also been shown to induce global increase in CTD-K7acetylation (*Schroeder et al., 2013*). To test whether the genes with highest K7me2/K7ac ratios are more sensitive to up-regulation upon TSA treatment, we mined published microarray data from mouse ES cells treated with TSA (*Karantzali et al., 2008*). We find that TSA has different effects on the three groups of genes

according to their K7me2/K7ac ratio (*Figure 7f*). The genes with high K7me2/K7ac ratio are more likely to be upregulated (107/141 genes) upon TSA treatment, than downregulated (34/141 genes). Genes in the medium ratio group are equally likely to be upregulated (51/91 genes) or downregulated (40/91 genes). Finally, genes with the lowest ratio are most often downregulated (142/180 genes) than upregulated (38/180 genes). The effect of TSA on three groups of genes is significantly different (Fisher's exact test; high versus low ratio: $p < 2.2 \times 10^{-16}$). We observed a similar trend when we treated our ES cell line with TSA; genes with higher ratios tend to be upregulated (*Figure 7—figure supplement 3*).

To test the converse effect of K7ac depletion on the levels of K7me, we treated cells with P300 inhibitor C646, which was previously shown to deplete K7ac levels (*Schroeder et al., 2013*). We find that inhibition of P300 induces an average 1.7-fold increase in the global levels of K7me2 (ranging from 1.2 to 2.5-fold across replicates), whereas the levels of total RPB1 remain constant or are slightly increased (*Figure 7g*). Levels of S7p also remain constant after P300 inhibition with C646 suggesting that K7ac and S7p contribute independently to productive elongation (*Figure 7g*), an observation which is consistent with the non-redundant contributions of K7ac and S7p to S2p and mRNA levels found by partial correlation and stepwise regression analyses (*Figure 6*).

Taken together, our results suggest that the extent of productive transcription at active genes with similar RNAPII recruitment to their promoters, is modulated at the level of CTD-K7 modifications, with methylation being associated with less productive transcription, and K7 acetylation being most enriched at the highest expressed genes.

## Discussion

The expansion of the RPB1 CTD in repeat number and amino acid diversity has been associated with increased complexity in specific groups of organisms, including vertebrates and plants (*Corden, 2013*; *Yang and Stiller, 2014*). Larger CTDs are thought to accommodate larger protein complexes. Non-canonical CTD repeats have the potential to expand the diversity of post-translational modifications, resulting in more complex signaling through the CTD. Although there is increasing recognition that complex CTDs may have specific roles in the regulation of more complex developmental programs, gene structures or gene organization (*Litingtung et al., 1999*; *Liu et al., 2010*; *Simonti et al., 2015*), our understanding of CTD signaling remains extremely limited, particularly as we have not yet identified all CTD modifications.

A small number of modifications to non-canonical residues have been identified, and their importance for the expression of specific subgroups of genes revealed. In mammalian cells, methylation of a single CTD-R7 is involved in expression of snRNAs and snoRNAs (*Sims et al., 2011*). Acetylation of CTD-K7 residues, the most common non-canonical substitution, has been associated with inducible gene expression of EGF-regulated genes (*Schroeder et al., 2013*).

In the present study, we show that CTD-K7 residues are mono- and di-methylated in mouse and human cells, and in the invertebrates *C. elegans* and *D. melanogaster*. Using murine cell lines with K7-to-S7 substitutions, we show that several CTD-K7 residues are simultaneously mono- and di-methylated, including a single K7 residue at repeat 35 of the murine CTD. Conservation of CTD-K7 methylation, and expansion of the number of methylated CTD-K7 residues from invertebrates to mammals, suggests functional relevance of this modification across taxa. Model systems characterized by K7 expansion (e.g. *Plasmodium* parasites; *Kishore et al., 2009*) further support the need for studying the extent of K7 modifications at RNAPII CTD.

We demonstrate that CTD-K7 mono- and di-methylation are associated with the earliest stages of transcription, are refractory to CDK9 inhibition using flavopiridol, and are distributed in discrete foci in the nucleoplasm of single cells. We mapped the genome-wide occupancy of K7me1 and K7me2 in mouse ES cells and find that they are highly enriched at the promoters of active genes. CTD-K7 methylation levels are highly correlated with promoter levels of unmodified S2, S7p, and S5p, but are uncoupled from mRNA expression and S2p abundance. In contrast, we find that CTD-K7 acetylation expands downstream of promoter regions into gene bodies, and highly correlates with S2p and mRNA levels.

One interesting aspect that emerges from studying CTD-K7 methylation and acetylation is that the two marks may compete for the eight available K7 residues at the CTD and thereby regulate gene expression levels. We observed that the ratio between K7me2 and K7ac at active promoters is

inversely related to gene expression. Genes with similar promoter occupancy of RNAPII but which have higher levels of K7me2 relative to K7ac are significantly less expressed than genes that have higher levels of K7ac relative to K7me2. This suggests that a balance between K7 modifications fine-tunes gene expression levels.

Further studies are required to explore the enzymatic activities that methylate and demethylate CTD-K7 residues, as well as the readers of methylated CTD-K7. We performed co-immunoprecipitation experiments of proteins associated with YFP-tagged RPB1 containing the wild-type CTD or the CTD bearing all K7-to-S7 mutations, followed by mass spectrometry, but this strategy failed to identify specific modifiers and readers (data not shown), suggesting that the interactions of RNAPII with CTD-K7 modifiers and/or readers may be transient and not easily captured by in vitro biochemical approaches.

The identification of modified CTD-K7 residues (K7me1, K7me2, K7me3 and K7ac) expands the repertoire of CTD interactors to proteins containing domains that recognize these marks, such as PHD finger, bromo-, chromo-, tudor- and MBT-domains (*Biggar and Li, 2015*; *Hamamoto et al., 2015*). Cross-talk between K7 modification and other CTD modifications, such as phosphorylation, may also regulate the extent of factor recruitment. Future investigations of the CTD code should consider the contribution of K7 modifications to co-transcriptional gene regulation in development and disease.

## Materials and methods

### Cell culture

Mouse ES cell line OS25 (generated in the laboratory of Austin G. Smith, and kindly donated by Wendy Bickmore) was grown on 0.1% gelatin-coated surfaces in GMEM-BHK21 supplemented with 10% fetal calf serum, 2 mM L-glutamine, 1% MEM non-essential amino acids, 1 mM sodium pyruvate, 50 μM β-mercaptoethanol (all from Gibco, Invitrogen; Waltham, MA), 1000 U/ml of human recombinant leukaemia inhibitory factor (LIF; Chemicon, Millipore; Germany) and 0.1 mg/ml Hygromycin (Roche; Switzerland), as described previously (*Billon et al., 2002*; *Niwa et al., 2000*). For inhibition of CDK9 activity and S2 phosphorylation, ES cells were treated for 1 hr with 10 μM of flavopiridol prior to protein extraction (*Stock et al., 2007*); from 50 mM stock in dimethyl sulfoxide (DMSO); a kind gift from Sanofi-Aventis, provided by Drug Synthesis and Chemistry Branch, Developmental Therapeutics Program, Division of Cancer Treatment and Diagnosis, National Cancer Institute, Bethesda, MD. For general deacetylase inhibition, ES cells were treated with TSA (50 nM, 3 hr); western blotting for H3K9ac confirmed efficient deacetylase inhibition (not shown). For inhibition of P300 acetylase, ES cells were treated (3h) with 30 μM C646 (Sigma, # SML0002; Germany) inhibitor, prior to protein extractions. For flavopiridol, TSA and C646 treatments, control cells were treated with vehicle (DMSO).

NIH-3T3 cells (kind donation from Alex Sardini) were grown in Dulbecco's modified Eagle's medium (DMEM) supplemented with 10% fetal calf serum, 2 mM L-glutamine and 1 mM sodium pyruvate (all from Gibco, Invitrogen). NIH-3T3 derived cell lines 8K, 3K, 1K and 0K were grown in presence of α-amanitin selection (2 μg/ml; Sigma).

Mouse ES cell line (OS25) and mouse NIH-3T3 fibroblast line were negative for mycoplasma, and routinely tested for mycoplasma with PCR mycoplasma test kit (AppliChem; Germany), according to manufacturer's instructions.

### Generation of NIH-3T3 cell lines expressing wild-type and mutant CTD constructs

The RPB1 construct *YFP-Rpb1 amr* contains a point mutation that gives resistance to α-amanitin (kind gift from Xavier Darzacq; *Darzacq et al., 2007*). The construct was digested with *SanDI* and *HpaI* (Thermo Fisher Scientific; Waltham, MA) to remove a fragment with 1.3 kb (residues 6126 to 7428) containing the *Rpb1* CTD. Different CTD sequences with wild-type (8K) and mutant repeats (3K, 1K and 0K) were obtained by gene synthesis (Genscript; Piscataway, NJ ) in the pUC57 vector backbone. In the mutant constructs, the K7 residues were converted into the consensus S7 residues. Mutant 3K contains only three of the K7 residues present at repeats 35, 40 and 47, mutant 1K contains only one K7 residue present at repeat 35 and mutant 0K has all K7 residues mutated to S7.

Wild-type and mutant CTDs were cloned from the pUC57 vectors to the *YFP-Rpb1 amr* backbone using the *SanDI* and *HpaI* restriction sites. Constructs were digested and sequenced to confirm the presence of the eight K7 codons at the wild-type construct, the K7 to S7 mutations at the different mutant contructs and the absence of additional sequence changes. For generation of cell lines 8K, 3K, 1K and 0K, NIH-3T3 cells growing in a 6-well dish were transfected with 3 µg of DNA using lipofectamine 2000 (Invitrogen, # 11668). α-Amanitin selection (2 µg/ml, Sigma) was started 24 hr after transfection. Polyclonal cell lines were kept under selection and expanded for 1 month before being used in the experiments described.

## Generation and purification of antibodies against CTD-K7me1 and CTD-K7me2

All handling of mice was approved by the Hokkaido University Animal Experiment Committee (approval number: 11-0109) and carried out according to guidelines for animal experimentation at Hokkaido University, where MAB Institute is located. Animals were housed in a designated pathogen-free facility at Hokkaido University. Mice were humanely euthanized via cervical dislocation by technically proficient individuals.

Mouse monoclonal antibodies were produced essentially as described previously (*Kimura et al., 2008*). Briefly, to generate monoclonal antibodies specific for CTD-K7me1, me2 and me3, mice were immunized with synthetic peptides (*Figure 3a*) that were conjugated with keyhole limpet hemocyanin. Hybridoma clones were generated and screened by ELISA using 96-well microtiter plates that were coated with the synthetic peptides (*Table 1*) conjugated with bovine serum albumin. Microtiter plates were incubated with 3-fold dilutions of each antibody starting from 1:20 dilution of a hybridoma culture supernatant, washed with phosphate-buffered saline (PBS), incubated with peroxidase-conjugated secondary antibody, and washed with PBS, before the colorimetric detection (450 nm absorbance) using tetramethylbenzidine. ELISA-positive clones were further validated using western blotting to attest their RNAPII specificity.

Hybridoma clones CMA611 and CMA612 used in this study were specific for K7me1 and K7me2 peptides, respectively. Prior to purification, CMA611 and CMA612 antibody clones were isotyped as IgG2b-κ and IgG1-κ, respectively, using a Mouse Monoclonal Antibody Isotyping Kit (AbD Serotec, #MMT1; Hercules, CA).

Clones were expanded in CD Hybridoma medium (Life Technologies) and culture supernatant was collected (for CMA612 clone NaCl was added to a final concentration of 4M), filtered using a 0.45 µm membrane (Nalgene; Rochester, NY) and purified using HiTrap Protein A FF Sepharose columns (1 ml; GE Healthcare; UK). Elution of CMA611 clone was performed using elution buffer (0.1 M glycine-HCl, pH 2.8). Elution of CMA612 clone was performed using mouse IgG1 mild elution buffer (Thermo Scientific). Antibody elution fractions were concentrated and the buffer was replaced to PBS using centrifugal filtration with Amicon Ultra filter 50K (Millipore).

## Western blotting

Whole cell extracts were collected by scraping cells in ice-cold 'lysis' buffer [50 mM Tris-HCl pH 7.5, 1 mM EDTA, 10% glycerol, 50 mM NaF, 5 mM sodium pyrophosphate, 1% Triton X-100 (all from Sigma), 1 mM DTT (Invitrogen), supplemented with phosphatase inhibitors $Na_3VO_4$ (2 mM) and PMSF (1 mM, Sigma) and protease inhibitor cocktail (Complete Mini EDTA-free, Roche # 11836170001)]. To release chromatin-associated proteins, cell extracts were sheared by passage through a 25G needle, as in *Stock et al., 2007*. Protein content of cell extracts was quantified using either Bradford (Bio-Rad, # 500-0205) or Bio-Rad DC protein assay (Bio-Rad, #500-0116). The following amounts of total extracts were used for western blotting: 0.5–2 µg total protein for 4H8 (S5p) and 8WG16 (unmodified-S2) antibodies, and 5–10 µg total protein for the other RNAPII antibodies/ modifications and 20 µg for histone H3K9 acetylation. Protein extracts were resolved on 3%–8% Tris-acetate sodium dodecyl sulfate polyacrylamide gel electrophoresis (SDS-PAGE) gels (Novex, Life Technologies; Waltham, MA) for RNAPII modifications and 15% SDS-PAGE gels for histone H3K9 acetylation. Membranes were blocked, incubated for primary antibody, washed and incubated for secondary antibody all in blocking buffer (10 mM Tris-HCl pH 8.0, 150 mM NaCl, 0.1% Tween-20, 5% non-fat dry milk; all from Sigma). Horseradish peroxidase (HRP)-conjugated secondary antibodies (Jackson ImmunoResearch Labs; Westgrove, PA) were detected with ECL western blotting

**Table 1.** RPB1 CTD peptides. CTD peptides with unmodified, mono-, di-, tri-methyl and acetyl K7 residues were used in ELISA assays to characterize the specificity of the CTD methyl antibodies produced in this study.

| Peptide sequence | Modification |
| --- | --- |
| SYSPTSP**K**YTPTSPSC | Unmodified |
| SYSPTSP**Kme1**YTPTSPSC | K7 monomethyl |
| SYSPTSP**Kme2**YTPTSPSC | K7 dimethyl |
| SYSPTSP**Kme3**YTPTSPSC | K7 trimethyl |
| SYSPTSP**Kac**YTPTSPSC | K7 acetyl |

CTD, C-terminal domain; ELISA, enzyme-linked immunosorbent assay.

detection reagents (Amersham, GE Healthcare). Detailed information about antibodies is shown in *Table 2*.

To remove phospho-epitopes from blots after protein transfer, membranes were treated with Alkaline Phosphatase Calf Intestinal (1 U/µl; NEB, #M0290S; Ipswich, MA) for 8 hr at 37°C in NEB buffer 3, prior to western blotting; control untreated blots were incubated with NEB buffer 3.

*C. elegans* whole worm extracts (kind donation from Stefanie Seelk and Baris Tursun) were prepared from N2 worms (wild-type). Worms were washed in M9 medium (42 mM $Na_2HPO_4$, 22 mM $KH_2PO_4$, 86 mM NaCl, 1 mM $MgSO_4$) and directly lysed in SDS sample buffer for 10 min at 94°C. Nuclei from *D. melanogaster* embryos (kind donation from Robert Zinzen) were isolated as previously described in *Bonn et al. (2012)*. Protein extracts were obtained using 'lysis' buffer as described above.

## Immunofluorescence

Mouse fibroblasts (NIH-3T3 cell line) were grown on coverslips coated with poly-L-lysine (Sigma). Cells were fixed at room temperature for 10 min in 4% freshly depolymerized paraformaldehyde, 0.1% Triton X-100 in 125 mM HEPES-NaOH (pH 7.6) and permeabilized in 1% Triton X-100 in PBS for 20 min at room temperature. After permeabilization, coverslips were washed in PBS and incubated with 20 mM glycine (Sigma) in PBS for 30 min at room temperature.

Blocking (1 hr), incubation with primary (2 hr) and secondary (1 hr) antibodies was performed in 'PBS plus' (PBS supplemented with 1% casein, 1% BSA, 0.2% fish skin gelatin, pH 7.8, all from Sigma). Washes were done using 'PBS plus' (after primary and secondary antibody incubations). Prior to nuclei acid staining, coverslips were washed with 0.1% Tween-20 in PBS. Nucleic acids were stained using 2 µM TOTO-3 iodide (Invitrogen) in 0.1% Tween-20 in PBS for 20 min at room temperature. Coverslips were mounted in VectaShield (Vector Laboratories, UK) and imaging was performed using a laser scanning confocal microscope Leica TCS SP2.

## Chromatin immunoprecipitation

ChIP for RNAPII modifications was performed using fixed chromatin as described previously (*Brookes et al., 2012*; *Stock et al., 2007*). Mouse ES cells were fixed in 1% formaldehyde (Sigma) at 37°C for 10 min after which the reaction was stopped adding glycine to a final concentration of 0.125M. Fixed cells were washed with ice cold PBS, lysed in 'swelling buffer' [25 mM HEPES pH 7.9, 1.5 mM $MgCl_2$, 10 mM KCl (all from Sigma) and 0.1% NP-40 (Roche)], scraped from dishes and nuclei were isolated with a Dounce homogenizer (tight pestle) followed by centrifugation. Nuclei resuspended ($1\times10^7$ nuclei/ml) in 'sonication' buffer [50 mM HEPES pH 7.9, 140 mM NaCl, 1 mM EDTA, 1% Triton X-100, 0.1% Na-deoxycholate and 0.1% SDS (all from Sigma)] were sonicated (Diagenode Bioruptor) for 30 min at full power for 30 cycles of 30 s "on" and 30 s "off" at 4°C. Chromatin was centrifuged and, after disposal of the insoluble fraction, DNA content was quantified using alkaline lysis. Both swelling and sonication buffers were supplemented with phosphatase inhibitors sodium flouride (5 mM), $Na_3VO_4$ (2 mM) and phenylmethylsulfonyl fluoride (PMSF; 1 mM) and protease inhibitor cocktail (Complete Mini EDTA-free, Roche # 11836170001).

**Table 2.** List of Antibodies used in this study. Full description of the antibodies and the amounts or concentrations used in this study for WB, ChIP or IF.

| Antibody | Raised in (isotype) | Clone | Stock | Amount/dilution | | | Source |
|---|---|---|---|---|---|---|---|
| | | | | WB | ChIP | IF | |
| S5p | Mouse (IgG) | CTD4H8 (MMS-128P) | 1 mg/ml | 1/200,000 | 10 µl (10 µg) | 1/3000 | Covance |
| S7p | Rat (IgG) | 4E12 | - | 1/10 | - | - | Kind gift from Dirk Eick |
| S2p | Mouse (IgM) | H5 (MMS-129R) | 1–3 mg/ml | 1/500 | - | - | Covance |
| Unphospho-S2 | Mouse (IgG) | 8WG16 (MMS-126R) | 1–3 mg/ml | 1/200 | - | - | Covance |
| N-terminus (Total RPB1) | Rabbit (IgG) | H224 (sc-9001x) | 200 µg/ml | 1/200 | - | - | Santa Cruz Biotechnology |
| K7me1 | Mouse (IgG) | CMA611 | 10 mg/ml | 1/1000 | 5 µl (50 µg) | 1/200 | This study |
| K7me2 | Mouse (IgG) | CMA612 | 10 mg/ml | 1/1000 | 5 µl (50 µg) | 1/200 | This study |
| H3K9ac | Rabbit serum | 39585 | - | 1/1000 | - | - | Active Motif |
| Lamin B | Goat (IgG) | C-20 Sc-6216 | 200 µg/ml | 1/500 | - | - | Santa Cruz Biotechnology |
| α-tubulin | Mouse (IgG) | T6074 | 2 mg/ml | 1/10,000 | - | - | Sigma |
| GFP | Rabbit (IgG) | A11122 | 2 mg/ml | 1/1000 | - | - | Life Technologies |
| Digoxigenin | Mouse (IgG) | 200–002-156 | 1.2 mg/ml | - | 10 µl (12 µg) | - | Jackson ImmunoResearch |

ChIP, chromatin immunoprecipitation; IF, immunofluorescence; WB, western blotting.

Protein G magnetic beads (Active Motif; Carlsbad, CA) were incubated (50 µl per IP) with 10 µg of bridging antibody anti-IgM/IgG (Jackson ImmunoResearch) for 1 hr at 4°C and then washed and resuspended in sonication buffer. For each immunoprecipitation, 600–700 µg of chromatin was incubated overnight with beads and the respective RNAPII or control antibody (for details about ChIP antibodies see *Table 2*). After immunoprecipitation, beads were washed at 4°C as described previously (*Stock et al., 2007*). For elution of the immune complexes, beads were resuspended in 50 mM Tris-HCl pH 8.0, 1 mM EDTA, 1% SDS and incubated for 5 min at 65°C, followed by 15 min at room temperature. Reverse cross-linking was done at 65°C overnight (for qPCR) or for 8 hr (for samples processed for chromatin immunoprecipitation with sequencing (ChIP-seq) library preparation) after adding 10 µg of RNase A (Sigma, #R4642) and NaCl to a final concentration of 155 mM.

Samples were incubated with 100 µg of Proteinase K (Roche, # 03115836001), after EDTA concentration was adjusted to 5 mM, for 2h at 50°C. DNA was recovered by phenol-chloroform extraction followed by ethanol precipitation in the presence of 20 ng/ml glycogen. DNA concentration was determined by PicoGreen fluorimetric assay (Molecular Probes, Invitrogen; Waltham, MA) and samples were diluted to a final concentration of 0.2 ng/µl. Immunoprecipitated and input samples (0.5 ng each) were analyzed by quantitative real-time PCR (qPCR) using SYBR No-Rox sensimix (Bioline, UK). Quantitative PCR 'cycle over threshold' (Ct) values from immunoprecipitated samples (RNAPII or control antibody) were subtracted from the input Ct values and the fold enrichment over input was calculated using the equation $2^{(\text{input Ct} - \text{IP Ct})}$. Primer sequences are listed in *Table 3*.

### ChIP-seq library preparation and sequencing

Prior to ChIP-seq library preparation, RNAPII enrichment and distribution were assessed by qPCR analyses using a previously characterized panel of genes (*Brookes et al., 2012*; *Stock et al., 2007*). ChIP-seq libraries for CTD-K7me1 and K7me2 were prepared from 10 ng of immunoprecipitated DNA (quantified by PicoGreen and Qubit) using the Next ChIP-Seq Library Prep Master Mix Set from Illumina (NEB, # E6240) following the NEB protocol, with some modifications. The intermediate products from the different steps of the NEB protocol were purified using MiniElute PCR purification

**Table 3.** List of PCR primers used in this study. Primer sequences (F, forward; R, reverse) are represented (5′to 3′orientation) for promoter and coding regions of active and inactive genes. Primers designed for the *Polr2a* locus (scheme in **Figure 5—figure supplement 1a**) cover the promoter region (–1 and –0.5 kb), exons (E1 and E28), intron (I1), exon-intron boundaries (I2/E3 and E19/I19) and downstream of TES (+2 kb TES).

| Gene | Primer sequence | Gene | Primer sequence |
|---|---|---|---|
| ChIP primers β-actin (promoter) F | GCAGGCCTAGTAACCGAGACA | Gene expression primers β-actin 5′F | CCACCCGCGAGCACA |
| β-actin (promoter) R | AGTTTTGGCGATGGGTGCT | β-actin 5′R | CCGGCGTCCCTGCTTAC |
| β-actin (coding) F | TCCTGGCCTCACTGTCCAC | β-actin Exon1 F | TCTTTGCAGCTCCTTCGTTG |
| β-actin (coding) R | GTCCGCCTAGAAGCACTTGC | β-actin Exon2 R | ACGATGGAGGGGAATACAGC |
| Oct4 (promoter) F | GGCTCTCCAGAGGATGGCTGAG | Oct4 5′F | TGAGCCGTCTTTCCACCA |
| Oct4 (promoter) R | TCGGATGCCCCATCGCA | Oct4 5′R | TGAGCCTGGTCCGATTCC |
| Oct4 (coding) F | CCTGCAGAAGGAGCTAGAACA | Sox2 5′F | AGGGCTGGGAGAAAGAAGAG |
| Oct4 (coding) R | TGTGGAGAAGCAGCTCCTAAG | Sox2 5′R | ATCTGGCGGAGAATAGTTGG |
| Myf5 (promoter) F | GGAGATCCGTGCGTTAAGAATCC | Serpine1 5′F | CCCCGAGAGCTTTGTGAAG |
| Myf5 (promoter) R | CGGTAGCAAGACATTAAAGTTCCGTA | Serpine1 5′R | AAGGTGCCTTGTGATTGGCT |
| Myf5 (coding) F | GATTGCTTGTCCAGCATTGT | Dusp1 5′F | CGGTGAAGCCAGATTAGGAG |
| Myf5 (coding) R | AGTGATCATCGGGAGAGAGTT | Dusp1 5′R | AGGAGCGACAATCCAACAAC |
| Gata1 (promoter) F | AGAGGAGGGGAGAAGGTGAGTG | Ctgf 5′F | GACTCAGCCAGATCCACTCC |
| Gata1 (promoter) R | AGCCACCTTAGTGGTATGACG | Ctgf 5′R | GTGCAGAGGCGACGAGAG |
| Gata1 (coding) F | TGGATTTTCCTGGTCTAGGG | | |
| Gata1 (coding) R | GTAGGCCTCAGCTTCTCTGTAGTA | | |
| Polr2a (-1 kb) F | CCGTAAAGCTATTAGAGCACAGG | | |
| Polr2a (-1 kb) R | ATGCATAAGGCAGGCAAGAT | | |
| Polr2a (-0.5 kb) F | GTAACCTCTGCCGTTCAGGA | | |
| Polr2a (-0.5 kb) R | TTTCTCCCTTTCCGGAGATT | | |
| Polr2a (E1) F | CAGGCTTTTTGTAGCGAGGT | | |
| Polr2a (E1) R | GACTCAGGACTCCGAACTGC | | |
| Polr2a (I1) F | CAGAGGGCTCTTTGAATTGG | | |
| Polr2a (I1) R | GCATCAGATCCCCTTCATGT | | |
| Polr2a (I2/E3) F | GCCCTCTTCTGGAGTGTCTG | | |
| Polr2a (I2/E3) R | AGGAAGCCCACATGAAACAC | | |
| Polr2a (E19/I19) F | CCAAGTTCAACCAAGCCATT | | |
| Polr2a (E19/I19) R | TCTTAACCGCTGAGCCATCT | | |
| Polr2a (E28) F | TCTCCCACTTCTCCTGGCTA | | |
| Polr2a (E28) R | CCGAGGTTGTCTGACCCTAA | | |
| Polr2a (+2 kb TES) F | GAGGGGCAGACACTACCAAA | | |
| Polr2a (+2 kb TES) R | AAAAGGCCAAAGGCAAAGAT | | |

kit (Qiagen, # 28004, Germany). Adaptors, PCR amplification primers and indexing primers were from the Multiplexing Sample Preparation Oligonucleotide Kit (Illumina, # PE-400–1001; San Diego, CA). Samples were PCR amplified prior to size selection (250–600 bp) on an agarose gel. After purification by QIAquick Gel Extraction kit (Qiagen, # 28704), libraries were quantified by qPCR using Kapa Library Quantification Universal Kit (KapaBiosystems, #KK4824; Wilmington, MA). Library size distribution was assessed by Bioanalyzer High Sensitivity DNA analysis Kit (Agilent, #5067–4626, Santa Clara, CA) before high-throughput sequencing. Libraries were quantified by Qubit and sequenced using Illumina Sequencing Technology (single-end sequencing, 51 nucleotides) at the

**Table 4.** Description of ChIP-seq and messenger RNA datasets used in this study. Full description of the ChIP-seq datasets produced or re-analysed in this study. NCBI Gene Expression Omnibus (GEO) Sample reference is indicated for published datasets.

| ChIP-seq dataset | Dataset origin | Antibody clone | Mapped reads (millions) | ES cell line |
|---|---|---|---|---|
| RPB1-K7me1 (GSM1874007) | This study | CMA611 (this study) | 64 | ESC OS25 |
| RPB1-K7me2 (GSM1874008) | This study | CMA612 (this study) | 69 | ESC OS25 |
| RPB1-K7ac (SRR1028808) | *Schröder et al. (2013)* | AcRPB1 (*Schröder et al., 2013*) | 85 | ESC |
| Input (SRR1028807) | *Schröder et al. (2013)* | - | 21 | ESC |
| RPB1-S5p (GSM850467) | *Brookes et al. (2012)* | CTD4H8 (MMS-128P, Covance) | 22 | ESC OS25 |
| RPB1-S7p (GSM850468) | *Brookes et al. (2012)* | 4E12 (*Chapman et al., 2007*) | 11 | ESC OS25 |
| RPB1-S2p (GSM850470) | *Brookes et al. (2012)* | H5 (MMS-129R, Covance) | 33 | ESC OS25 |
| Unphospho-S2 (8WG16) (GSM850469) | *Brookes et al. (2012)* | 8WG16 (MMS-126R, Covance) | 24 | ESC OS25 |
| Mock IP (GSM850473) | *Brookes et al. (2012)* | - | 12 | ESC OS25 |
| H3K4me3 (GSM307618) | *Mikkelsen et al. (2007)* | ab8580 (Abcam) | 9 | ESC V6.5 |
| H3K36me3 (GSM850472) | *Brookes et al. (2012)* | 13C9 (*Rechtsteiner et al., 2010*) | 23 | ESC OS25 |
| H2Aub1 (GSM850471) | *Brookes et al. (2012)* | E6C5 (Upstate) | 18 | ESC OS25 |
| H3K27me3 (GSM307619) | *Mikkelsen et al. (2007)* | 07–449 (Upstate) | 8 | ESC V6.5 |
| RNA datasets | Dataset origin | Mapped reads (millions) | | Cell line |
| mRNA-seq (GSM850476) | *Brookes et al. (2012)* | 74 | | ESC OS25 |
| GRO-seq (GSE48895) | *Jonkers et al. (2014)* | 25 | | ESC V6.5 ("untreated") |

ChIP-seq, chromatin immunoprecipitation with sequencing; ES, embryonic stem; GRO-seq, global run-on sequencing.

BIMSB Genomics Platform using an Illumina HiSeq2000, according to the manufacturer's instructions.

## Bioinformatics analyses

In addition to ChIP-seq datasets generated for CTD-K7me1 and K7me2 modifications in mouse ES cells, we also analyzed published ES cell ChIP-seq and GRO-seq datasets (*Table 4*). Published RNA-PII datasets were: unmodified-S2 (detected with antibody 8WG16), S5p, S7p, S2p, all from Brookes *et al.*, 2012 and RNAPII K7ac from *Schroeder et al., 2013*. Published histone modification datasets were: H2Aub1 and H3K36me3 from *Brookes et al., 2012*; and H3K27me3 and H3K4me3 from *Mikkelsen et al., 2007*. CpG content was defined as in *Brookes et al., 2012*.

Sequenced reads were aligned to the mouse genome (assembly mm9, July 2007) using Bowtie2 version 2.0.5 (*Langmead and Salzberg, 2012*), with default parameters. Duplicated reads (i.e. identical reads, aligned to the same genomic location) occurring more often than a threshold were removed. The threshold is computed for each dataset as the 95th percentile of the frequency distribution of reads.

Boxplots were produced using R. A pseudo-count of $10^{-4}$ was added to FPKM values from *Brookes et al., 2012* prior to logarithmic transformation and plotting.

Average ChIP-seq profiles were generated by plotting the average depth of coverage in non-overlapping windows of 10 bp, across 5 kb genomic windows centered on TSS and TES as in (*Brookes et al., 2012*). The read coverage of ChIP-seq heatmaps was calculated using HTSeq

(*Anders et al., 2015*) with 5 bp resolution. The z-score of each gene (row) was plotted using the pheatmap package (*Kolde, 2015*) in R.

GRO-seq data from *Jonkers et al., 2014* were downloaded as bedgraph files for untreated ES cells and the read coverage for sense and anti-sense transcription was calculated separately for a 1.5 kb window (–500 to +1000 bp of TSS) at a 10 bp resolution. GRO-seq RPKM (reads per kilobase per million of reads mapped) values were estimated using the total number of reads mapped from the TSS to the TES of genes.

For CTD-K7me1, K7me2 and K7ac, positively enriched windows were detected using BCP (*Xing et al., 2012*) run in Histone Mark (HM) mode using as control datasets: (a) the mock ChIP data-set from *Brookes et al., 2012* for CTD-K7me1 and CTD-K7me2, or (b) the input dataset from *Schroeder et al., 2013* for CTD-K7ac. Gene promoters were considered positive for K7me1, K7me2 and K7ac when (a) the 2 kb windows centered on the gene promoters coincided with a region enriched for the mark and (b) the amount of reads in the promoter window was above the 10th percentile (10% tail cut; *Figure 5—figure supplement 1b*). Genes whose (positive) TSS window overlapped other positive windows for the same mark were removed. Positive genes that were inside other positive genes for the same mark were also removed (only 'internal' gene removed). Excluded genes were classified as NA. For other RNAPII modifications or histone marks, we used published classification from (*Brookes et al., 2012*).

For the analyses in *Figure 7* and *Figure 7—figure supplement 2a*, we defined two cohorts of non-overlapping active genes. The larger group (n = 4271) included all genes with non-overlapping promoters (2 kb window around TSS), active, i.e. FPKM >1 (FPKM values from *Brookes et al., 2012*) and positive for S5p (TSS), S7p (TSS), S2p (2 kb window after TES), as well as negative for the histone marks H3K27me3 and H2AK119ub1. The smaller cohort (n = 1564) had the additional criterion that the maximum peak of RNAPII (8WG16) has to be within a 100 bp window centered at the TSS.

Genes classified as PRCr were positively marked by H3K27me3, H2AK119ub1 and RNAPII-S5p, and negatively marked by 8WG16, all at the 2 kb window centered around the genes TSS, and were devoid of RNAPII-S2p in the 2 kb window downstream of TES. They also do not overlap with other positively marked TSS or TES regions (i.e. different from classification 'NA' in *Brookes et al., 2012*).

Genes classified as most active (top 15%) or least active (bottom 15%) were defined according to FPKM values from mRNA-seq datasets published in (*Brookes et al., 2012*), and were chosen from the pool of genes negative for Polycomb marks (H2AK119ub1 and H3K27me3; classification from *Brookes et al., 2012*).

## Ratios of K7me1/K7ac and K7me2/K7ac

To relate CTD-K7 methylation to acetylation levels, we computed the ratio between the read counts of K7me1/2 and K7ac at promoters (2 kb around TSS), both scaled by 10 million over the total number of non-duplicated mapped reads of the respective dataset. The ratios K7me1/K7ac and K7me2/K7ac are highly correlated (Spearman's correlation coefficient 0.88); for simplicity, only the K7me2/K7ac ratio was used for further in-depth analyses. The ratio of K7me2/K7ac for each active gene was used to define three quantiles (high, medium and low K7me2/K7ac ratio).

## Correlations and linear modeling of CTD modifications

To apply linear regression models and perform correlation analyses, the promoter read counts for the different CTD modifications were centered and scaled. The Spearman's correlation coefficients were plotted as a matrix using the "corrplot" package in R and partial correlations calculated using the "pcor" function of "ggm" package. Traditional regression modeling using correlated predictor variables is difficult to interpret and hence we used a stepwise regression approach to better untangle the contributions of different CTD modifications. Typically, for a given number of predictor variables, we report the top five best models. Since CTD-S2p measured at 2 kb window after TES is the best predictor of mRNA amongst all CTD modifications and gene regions, we decided to study models that predict the levels of the S2p (instead of mRNA-seq) to avoid inclusion of another highly correlated variable. Model fitting was performed in R using the packages "lm" and "leaps". Model ranking was based on both adjusted $R^2$ and Mallow's $C_p$.

To investigate whether the contribution of CTD modifications to S2p levels at 2 kb window after the TES, which were obtained through correlation and stepwise regression analyses, are due to co-

linearity artifacts or over-fitting, we also employed the LASSO regression method implemented in the "lars" R package (*Efron, 2013*). We used as predictor variables the promoter levels of 8WG16, S5p, S7p, K7me1, K7me2, K7ac, mock and CpG. A typical LASSO path is shown in *Figure 6—figure supplement 2*. We used cross-validation [minimum cross-validated mean standard error + 1 standard error] to select the optimal position along the path; typical models discard the mock and 8WG16 predictors. As seen in stepwise regression, S7p and K7ac display significant positive coefficients while K7me1 and K7me2 have negative coefficients.

## Gene ontology analysis

Gene ontology (GO) enrichment analysis was performed using GO-Elite version 1.2.5 (Gladstone Institutes; http://genmapp.org/go_elite). Over-representation analysis of the top and bottom 300 genes for K7me2/K7ac ratio (see above) was performed using as background the n = 1564 active, non-overlapping genes for which the ratio was computed. Default parameters were used as filters: z-score threshold more than 1.96, *P*-value less than 0.05, number of genes changed more than 2. Over-representation analysis was performed with "permute p-value" option, 2000 permutations.

## Microarray data analysis

Total RNA transcriptomic analyses of 50 nM TSA treatment of mouse ES cells (6 hr) is publicly available and described in *Karantzali et al., 2008*. Affimetrix Mouse430 2.0 Array probeset names were converted to Ensembl Gene ID using the "Mouse430_2.na34.annot.csv" table obtained from the Affymetrix website. Probesets associated with more than one Ensembl Gene ID were excluded from the analysis; in the few cases where more than one probeset was associated to the same Ensembl Gene ID, one probe set was randomly selected. Merging of the list of TSA differentially expressed genes with the K7me2/K7ac ratio lists was performed using Ensembl Gene ID; genes in High, Medium or Low group were taken from extended cohort (4271 genes; see Bioinformatics analyses above). The significance of the difference between genes in the three ratio groups was tested with a two-tailed Fisher's exact test using R.

## Gene expression analysis by quantitative RT-PCR

RNA was extracted by Trizol (Ambion; Waltham, MA) extraction using Phase Lock Gel tubes (5 PRIME; Gaithersburg, MD). Samples were DNAse I (Turbo DNAse, Ambion) treated according to manufacturer instructions. Total RNA (1 μg) was retrotranscribed using random primers (50 ng) and 200 U of reverse transcriptase (SuperScript II RT, Invitrogen) in a 20 μl reaction. The synthesized complementary DNA was treated with 2 U of RNAse H (NEB), diluted 1:10 and 2.5 μl were used per RT-PCR reaction. Total RNA was measured using primers designed for the 5´end of genes and levels were normalized for *Actb* mRNA measured using primers for the exon1-exon2 junction (for primer sequence see *Table 3*).

## Acknowledgements

We thank Naohito Nozaki (MAB Institute) for producing the hybridoma clones, Robert Zinzen, Baris Tursun, Stefanie Seelk, Xavier Darzacq, Alex Sardini, Marina Chekulaeva, Marta Mauri, Francis Stewart, Andrea Kranz, Giovanni Ciotta and Dirk Eick for protein extracts, reagents or published antibodies, Guido Mastrobuoni, Stefan Kempa, Oliver Popp, and Gunnar Dittmar for mass spec analyses, Sanofi-Aventis and the National Cancer Institute (NIH) for generously providing flavopiridol, Enrique Martinez-Perez, Niall Dillon, Zoe Webster and Laurence Game for advice, the BIMSB Genomics Platform for Illumina sequencing, and Inês de Castro for the design of *Rpb1* primers. We thank Dirk Eick for discussion of unpublished results. JDD, CF, MC, EB, AP thank the MRC for support. JDD, TR, ETT, CF, AK and AP thank the Helmholtz Association for support. JDD was supported by Fundação para a Ciência e Tecnologia (grant - SFRH / BD / 51005 / 2010, Portugal). HK was supported by Grants-in-aid from the Japan Society for the Promotion of Science (25116005; 26291071) and the Core Research for Evolutional Science and Technology from Japan Science and Technology Agency.

## Additional information

### Funding

| Funder | Grant reference number | Author |
| --- | --- | --- |
| Medical Research Council | | João D Dias<br>Carmelo Ferrai<br>Mita Chotalia<br>Emily Brookes<br>Ana Pombo |
| Helmholtz-Gemeinschaft | | João D Dias<br>Tiago Rito<br>Elena Torlai Triglia<br>Alexander Kukalev<br>Carmelo Ferrai<br>Ana Pombo |
| Japan Society for the Promotion of Science | 25116005; 26291071 | Hiroshi Kimura |
| Japan Science and Technology Agency | | Hiroshi Kimura |
| Fundação para a Ciência e a Tecnologia | SFRH/BD/51005/2010 | João D Dias |

The funders had no role in study design, data collection and interpretation, or the decision to submit the work for publication.

### Author contributions

JDD, Final approval of the version to be published, Design of the project, Planning and execution of most experiments, Acquisition of data, Analysis and interpretation of data, Writing of the article, Drafting or revising the article, Contributed unpublished essential data or reagents; TR, Final approval of the version to be published, Computational analysis and interpretation of data, Revising the article, Conception and design; ETT, Final approval of the version to be published, Computational analysis and interpretation of data, Revising the article; AK, Final approval of the version to be published, Revising the article, ChIP-seq library preparation, Acquisition of data, Analysis and interpretation of data, Contributed unpublished essential data or reagents; CF, Final approval of the version to be published, Revising the article, ChIP-seq library preparation, Acquisition of data, Contributed unpublished essential data or reagents; MC, Final approval of the version to be published, Revising the article, Design of mutant constructs, Conception and design; EB, Final approval of the version to be published, Revising the article, Conception and design, Analysis and interpretation of data; HK, Final approval of the version to be published, Revising the article, Production of CTD-K7 methyl antibodies, Planning of experiments, Acquisition of data, Conception and design, Analysis and interpretation of data, Contributed unpublished essential data or reagents; AP, Final approval of the version to be published, Design of the project, Planning of experiments, Analysis and interpretation of data, Writing of the article, Conception and design, Drafting or revising the article

### Author ORCIDs

Ana Pombo, http://orcid.org/0000-0002-7493-6288

### Ethics

Animal experimentation: All handling of mice was approved by the Hokkaido University Animal Experiment Committee (approval number: 11-0109) and carried out according to guidelines for animal experimentation at Hokkaido University, where MAB Institute Inc. is located. Animals were housed in a designated pathogen-free facility at Hokkaido University. Mice were humanely euthanized via cervical dislocation by technically proficient individuals.

# Additional files

## Major datasets

The following datasets were generated:

| Author(s) | Year | Dataset title | Dataset URL | Database, license, and accessibility information |
|---|---|---|---|---|
| Dias JD, Rito T, Torlai Triglia E, Ferrai C, Kukalev A, Kimura H, Pombo A | 2015 | CTD-K7me1 ChIP-seq | http://www.ncbi.nlm.nih.gov/geo/query/acc.cgi?acc=GSM1874007 | Publicly available at the NCBI Gene Expression Omnibus (Accession no: GSM1874007). |
| Dias JD, Rito T, Torlai Triglia E, Ferrai C, Kukalev A, Kimura H, Pombo A | 2015 | CTD-K7me2 ChIP-seq | http://www.ncbi.nlm.nih.gov/geo/query/acc.cgi?acc=GSM1874008 | Publicly available at the NCBI Gene Expression Omnibus (Accession no: GSM1874008). |
| Dias JD, Rito T, Torlai Triglia E, Ferrai C, Kukalev A, Kimura H, Pombo A | 2015 | Expanding the CTD code: methylation of non-consensus Lysine residues marks early transcription in mammalian cells | http://www.ncbi.nlm.nih.gov/geo/query/acc.cgi?acc=GSE72876 | Publicly available at the NCBI Gene Expression Omnibus (Accession no: GSE72876). |

The following previously published datasets were used:

| Author(s) | Year | Dataset title | Dataset URL | Database, license, and accessibility information |
|---|---|---|---|---|
| Schröeder S, Herker E, Itzen F, He D, Thomas S, Gilchrist DA, Kaehlcke K, Cho S, Pollard KS, Capra JA, Schnoelzer M, Cole PA, Geyer M, Bruneau BG, Adelman K, Ott M | 2013 | acetyl PolII ChIP | https://trace.ddbj.nig.ac.jp/DRASearch/run?acc=SRR1028808 | Publicly available at the DNA Data Bank of Japan (Accession no: SRR1028808). |
| Schröeder S, Herker E, Itzen F, He D, Thomas S, Gilchrist DA, Kaehlcke K, Cho S, Pollard KS, Capra JA, Schnoelzer M, Cole PA, Geyer M, Bruneau BG, Adelman K, Ott M | 2013 | acetyl inputChromatin ChIP | https://trace.ddbj.nig.ac.jp/DRASearch/run?acc=SRR1028807 | Publicly available at the DNA Data Bank of Japan (Accession no: SRR1028807). |
| Brookes E, de Santiago I, Hebenstreit D, Morris KJ, Carroll T, Xie SQ, Stock JK, Heidemann M, Eick D, Nozaki N, Kimura H, Ragoussis J, Teichmann SA, Pombo A | 2012 | RNAPII S5P ChIPSeq | http://www.ncbi.nlm.ih.gov/geo/query/acc.cgi?acc=GSM850467 | Publicly available at the NCBI Gene Expression Omnibus (Accession no: GM850467). |
| Brookes E, de Santiago I, Hebenstreit D, Morris KJ, Carroll T, Xie SQ, Stock JK, Heidemann M, Eick D, Nozaki N, Kimura H, Ragoussis J, Teichmann SA, Pombo A | 2012 | RNAPII S7P ChIPSeq | http://www.ncbi.nlm.nih.gov/geo/query/acc.cgi?acc=GSM850468 | Publicly available at the NCBI Gene Expression Omnibus (Accession no: GSM850468). |

| | | | | |
|---|---|---|---|---|
| Brookes E, de Santiago I, Hebenstreit D, Morris KJ, Carroll T, Xie SQ, Stock JK, Heidemann M, Eick D, Nozaki N, Kimura H, Ragoussis J, Teichmann SA, Pombo A | 2012 | RNAPII S2P ChIPSeq | http://www.ncbi.nlm.nih.gov/geo/query/acc.cgi?acc=GSM850470 | Publicly available at the NCBI Gene Expression Omnibus (Accession no: GSM850470). |
| Brookes E, de Santiago I, Hebenstreit D, Morris KJ, Carroll T, Xie SQ, Stock JK, Heidemann M, Eick D, Nozaki N, Kimura H, Ragoussis J, Teichmann SA, Pombo A | 2012 | RNAPII 8WG16 ChIPSeq | http://www.ncbi.nlm.nih.gov/geo/query/acc.cgi?acc=GSM850469 | Publicly available at the NCBI Gene Expression Omnibus (Accession no: GSM850469). |
| Brookes E, de Santiago I, Hebenstreit D, Morris KJ, Carroll T, Xie SQ, Stock JK, Heidemann M, Eick D, Nozaki N, Kimura H, Ragoussis J, Teichmann SA, Pombo A | 2012 | Control MockIP | http://www.ncbi.nlm.nih.gov/geo/query/acc.cgi?acc=GSM850473 | Publicly available at the NCBI Gene Expression Omnibus (Accession no: GSM850473). |
| Brookes E, de Santiago I, Hebenstreit D, Morris KJ, Carroll T, Xie SQ, Stock JK, Heidemann M, Eick D, Nozaki N, Kimura H, Ragoussis J, Teichmann SA, Pombo A | 2012 | H3K36me3 ChIPSeq | http://www.ncbi.nlm.nih.gov/geo/query/acc.cgi?acc=GSM850472 | Publicly available at the NCBI Gene Expression Omnibus (Accession no: GSM850472). |
| Brookes E, de Santiago I, Hebenstreit D, Morris KJ, Carroll T, Xie SQ, Stock JK, Heidemann M, Eick D, Nozaki N, Kimura H, Ragoussis J, Teichmann SA, Pombo A | 2012 | H2Aub1 ChIPSeq | http://www.ncbi.nlm.nih.gov/geo/query/acc.cgi?acc=GSM850471 | Publicly available at the NCBI Gene Expression Omnibus (Accession no: GSM850471). |
| Brookes E, de Santiago I, Hebenstreit D, Morris KJ, Carroll T, Xie SQ, Stock JK, Heidemann M, Eick D, Nozaki N, Kimura H, Ragoussis J, Teichmann SA, Pombo A | 2012 | OS25 cells mRNA-Seq | http://www.ncbi.nlm.nih.gov/geo/query/acc.cgi?acc=GSM850476 | Publicly available at the NCBI Gene Expression Omnibus (Accession no: GSM850476). |
| Mikkelsen TS, Ku M, Jaffe DB, Issac B, Lieberman E, Giannoukos G, Alvarez P, Brockman W, Kim TK, Koche RP, Lee W, Mendenhall E, O'Donovan A, Presser A, Russ C, Xie X, Meissner A, Wernig M, Jaenisch R, Nusbaum C, Lander ES, Bernstein BE | 2007 | ES_H3K4me3_ChIPSeq | http://www.ncbi.nlm.nih.gov/geo/query/acc.cgi?acc=GSM307618 | Publicly available at the NCBI Gene Expression Omnibus (Accession no: GSM307618). |

| Mikkelsen TS, Ku M, Jaffe DB, Issac B, Lieberman E, Giannoukos G, Alvarez P, Brockman W, Kim TK, Koche RP, Lee W, Mendenhall E, O'Donovan A, Presser A, Russ C, Xie X, Meissner A, Wernig M, Jaenisch R, Nusbaum C, Lander ES, Bernstein BE | 2007 | ES_H3K27me3_ChIPSeq | http://www.ncbi.nlm.nih.gov/geo/query/acc.cgi?acc=GSM307619 | Publicly available at the NCBI Gene Expression Omnibus (Accession no: GSM307619). |
| Jonkers I, Kwak H, Lis JT | 2014 | V6.5_untreated_#1 | http://www.ncbi.nlm.nih.gov/geo/query/acc.cgi?acc=GSM1186440 | Publicly available at the NCBI Gene Expression Omnibus (Accession no: GSE48895). |

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
