## [Decision Letter]

Thank you for submitting your work entitled "Methylation of RNA polymerase II non-consensus Lysine residues marks early transcription in mammalian cells" for consideration by *eLife*. Your article has been reviewed by two peer reviewers, and the evaluation has been overseen, and independently reviewed, by a Reviewing Editor and Jim Kadonaga as the Senior Editor.

The reviewers have discussed the reviews with one another and the Reviewing editor has drafted this decision to help you prepare a revised submission.

Summary:

This study describes the characterisation of non-consensus CTD heptad repeat K7 residues on the C terminal portion of the RPB1 CTD. These lysine residues were previously shown to be subject to acetylation correlating with TSS associated pausing of EGF inducible genes. This study now shows that these same lysines can also be mono- or dimethylated based on the development of specific antibodies to these CTD modifications as well as the construction of variant CTDs with the 8 K7 reverted to consensus S7. Notably this S7 reverted mutant CTD still gives full cell viability using the α-amanitin resistant replacement construct in NIH-3T3 cells. This argues against a dominant function for the K7 heptads in these cells. Even so K7me1/2 clearly are present on promoters of active genes in ES cells based on ChIP-seq and this association is mainly unaffected by flavopiridol treatment arguing for their role in transcription initiation of many genes. Interestingly the K7acetyl CTD appears to also be involved in transcription elongation while K7me1/2 may be associated with the down-regulation of some genes. They also show that the K7me1/2 antibody isn't blocked by adjacent CTD serine phosphorylation, critical to allow the use of this antibody reagent.

Overall, it is the opinion of the editors and reviewers that this study is interesting and appropriate for publication in *eLife*, provided that additional experiments can be conducted to better establish the biological function of the K7 me1/2 marks and especially their interplay with K7 acetyl marks. In particular, we suggest the following experiments to improve your study.

Essential revisions:

1) To further study the interplay between K7 acetyl and me1/2 marks, a careful analysis of the potential switch in these marks following gene induction should be revealing.

2) Inhibition of HATs (i.e. K_d_ of P300 or use of chemical HAT inhibitors) to deplete K7Acetyl marks may increase K7 me1/2 and thereby cause gene repression. This should be tested.

3) Correlating K7me1/2 with low activity genes needs to be tested by looking at nascent RNA as well as steady state RNA. Looking at GR0-seq or chromatin-seq RNA to focus on transcription rather than mRNA levels would be informative.

4) Some understanding of which factors recognise these CTD K7me1/2 marks is needed. For example IP followed by mass spectrometry could be informative. It could also be tested whether SET1, which is recruited by CTD S5P to methylate histone H3K4 at promoters, also methylates CTD K7me1/2.

We recommend that you extend your analysis as indicated above. You should also be careful not to oversell the importance of K7me1/2 versus the dominant S2P and S5P marks. Use of "correlate" rather than "depend" might be better unless more definitive data is obtained.

---

## [Author Response]

Essential revisions: 1) To further study the interplay between K7 acetyl and me1/2 marks, a careful analysis of the potential switch in these marks following gene induction should be revealing.

We thank the reviewers for encouraging us to further investigate the interplay between K7me and K7ac in the context of gene expression changes. Without knowledge of the enzymatic activities that modulate K7me directly, we consulted published data about inhibitors that could modulate K7me/K7ac ratio, specifically Trichostatin A (TSA), previously shown to up-regulate global levels of K7ac (described in Schroeder et al., 2013).

We took advantage of published microarray data for mouse ES cells treated with the histone deacetylase inhibitor TSA (Karantzali et al., 2008). In light of our suggestion that the K7me2/K7ac relates with the potential for gene expression, we hypothesized that genes with the highest K7me2/K7ac ratios would be most sensitive to the TSA-induced global increase in K7ac, and therefore more likely to be up-regulated upon TSA treatment.

Integrating data from Karantzali et al. (2008) with our ChIP-seq data defining K7me-bound genes, we find that TSA has different effects on genes in the three K7me2/K7ac groups, with genes with a High K7me2/K7ac ratio being more likely to be up-regulated after TSA, in keeping with our hypothesis. This result is consistent with the observed relationship between K7me2/K7ac ratio and the extent of productive transcription at active genes, independently of the amount of RNAPII at gene promoters. We have included these results into the main text and added a new plot in Figure 7. The same observation was confirmed by RT-PCR for a small group of genes in our stem cell line using a shorter time of inhibition (3h) to minimize indirect effects (new Figure 7—figure supplement 3).

*2) Inhibition of HATs (i.e. K_d_ of P300 or use of chemical HAT inhibitors) to deplete K7Acetyl marks may increase K7 me1/2 and thereby cause gene repression. This should be tested.*

We thank the reviewers for the suggestion to interfere with P300, an acetyltransferase of CTDK7 residues, to further explore the interplay between CTD-K7 methylation and acetylation. The P300 inhibitor C646 reduces global CTD-K7 acetylation levels (Schroeder et al., 2013).

To examine the effects of the reduction of K7ac, we treated mouse ES cells with C646 (3h, 30 μM). We first confirmed the efficiency of P300 inhibition by analyzing H3K9ac levels, and found that it decreases, as expected (Bowers E.M. et al., 2010 Chem. Biol. 17, 471-482). We then analyzed global levels of CTD-K7 methylation by western blotting, and found a reproducible increase in CTD-K7me2 after inhibition of P300 (ranging between 1.2-2.5 fold across 2 biological and 6 technical replicates). This result combined with the observation that K7ac decreases (Schroeder et al., 2013) points to a dynamic balance between methylation and acetylation at CTD-K7 residues, which supports our interpretation of the genome-wide analyses of CTD marks.

The results of P300 inhibition on total RPB1, K7me2 and S7p were integrated in the manuscript, in Figure 7 and associated text. We have also tested the effect of C646 on K7me1 in a single biological replicate, and find that K7me1 levels are also increased (Figure 8).

Author response image 1.P300 inhibition promotes a small increase in global levels of CTD-K7me2.Mouse ES cells were treated with P300 inhibitor C646 (30 μM, 3h), before western blotting with antibodies specific for total RPB1, K7me1, K7me2 and S7p. Hypo- (IIa) and hyperphosphorylated (II0) RPB1 forms are indicated. Loading control: Lamin B.**DOI:**
http://dx.doi.org/10.7554/eLife.11215.025

To test the effects of P300 inhibition on genes with different K7me/K7ac ratios, we performed RT-PCR on 6 genes with a range of K7me/K7ac ratios; Figure 9). Interestingly, irrespectively of ratio, we observe a decrease in gene expression upon P300 inhibition in 4/6 genes tested. This suggests that irrespective of the K7ac level, its loss often results in gene downregulation (Figure 9). Alternatively, P300 inhibition may have other confounding effects, such as preventing histone acetylation and thereby repressing gene expression irrespectively of the K7me/ac ratio. We would prefer not to include in the manuscript the single gene expression results from C646 treated cells because we feel we cannot fully interpret the effects based on this small number of genes and potential confounding changes of histone acetylation levels.

Author response image 2.Down-regulation of gene expression after P300 inhibition.(**A**) Range of K7me2/K7ac ratios for genes used in expression analysis after C646 treatment. (**B**) Total RNA levels were measured by quantitative RT-PCR after treatment of ES cells with C646 (30 μM, 3h) or vehicle DMSO (control cells). Expression relative to control cells is represented and genes are ordered according to K7me2/K7ac ratio. Total RNA levels were measured using primers for the 5´end of each gene and normalized for *Actb* mRNA levels. Mean and standard deviations from 3 independent TSA treatments are represented.**DOI:**
http://dx.doi.org/10.7554/eLife.11215.026

To further explore the effect of K7ac loss (and K7me gain) on gene expression states, we also investigated how P300 inhibition impacts on S7p levels (Figure 8 and Figure 7). This experiment was inspired by the observations from the partial correlation analyses and stepwise regression that these marks have independent contributions to productive elongation. Interestingly, we observe that S7p levels do not change upon P300 inhibition, reinforcing the view that S7 phosphorylation is independent of K7 acetylation or methylation. This observation is consistent with our suggestion that both K7ac and S7 phosphorylation contribute independently to drive S2p levels, and points to a complex regulatory cascade that integrates K7 modifications with phosphorylation of serine residues. We agree with the reviewers that further studies will be necessary to fully understand the functional roles of K7me2 and K7me1 in gene regulation, and feel that the revised manuscript provides the starting point for in-depth dissection of such mechanisms.

3) Correlating K7me1/2 with low activity genes needs to be tested by looking at nascent RNA as well as steady state RNA. Looking at GR0-seq or chromatin-seq RNA to focus on transcription rather than mRNA levels would be informative.

We thank the reviewers for prompting us to investigate how the ratio between K7 methylation/acetylation relates with nascent RNA. We have analyzed nascent transcript data obtained from mouse ES cells, using published Global Run-On sequencing (GRO-seq) data from Jonkers et al. (2014). In agreement with the results obtained for mRNA-seq data (Figure 7), we find that the K7me2/K7ac ratio is also anti-correlated with the amount of nascent transcription (Spearman’s correlation coefficient = – 0.31, p-value < 2.2x10-16). These results reinforce the finding that K7 methylation is an early transcription mark and that the balance between K7 modifications helps fine-tune transcription. The GRO-seq levels across K7me/K7ac ratios have been included in an extra panel in Figure 7. Additionally, a boxplot summarizing GRO-seq values for the different K7 ratio quantiles and heatmaps of GRO-seq signal have been added to Figure 7—figure supplement 1 and Figure 7—figure supplement 2.

*4) Some understanding of which factors recognise these CTD K7me1/2 marks is needed. For example IP followed by mass spectrometry could be informative.*

We agree with the reviewers that identifying readers of CTD K7me1/2 marks is a most pressing task, which we have tried in various ways. As alluded in the Discussion of the first version of the manuscript, we attempted to identify potential readers of K7 methylation, by performing mass spectrometry after immunoprecipitation of YFP-tagged wild-type (8K) RPB1 versus 0K mutant. However, this approach did not yield reproducible protein identifications in biological replicates, probably due to the fact that K7 methylation might be associated with a very small pool of RNAPII complexes found at the earliest stages of transcription. The staining of only small and discrete dots by IF (Figure 4) supports the notion that only a fraction of RNAPII complexes might be heavily marked by K7 methylation. Moreover, in such a global RNAPII pull-down, any differences in protein recruitment due to a specific CTD modification may be extensively diluted by all the proteins that bind to RNAPII abundant Serine modifications, irrespective of K7 modification.

In response to the reviewers’ requests we attempted a peptide pull-down approach, where we designed a two-step peptide pull-down strategy to identify possible factors that recognise CTDK7me2, but not unmodified K7 peptides. Total protein extracts from mouse ES cells were first incubated with bead-immobilized peptides comprising unmodified non-canonical CTD sequence (3 heptapeptide repeats Y_1_-S_2_-P_3_-T_4_-S_5_-P_6_-K_7_). We hypothesized that this step should clear up mouse ES cell extracts from proteins that bind to non-canonical CTD sequence in the absence of K7 modifications. After this pre-clearing step, we performed pull-downs using beads carrying dimethylated K7 peptides (3 heptapeptide repeats Y_1_-S_2_-P_3_-T_4_-S_5_-P_6_-K7me2). Proteins obtained after the first and second pull-downs (two biological replicates each) were analyzed by LS-MS/MS.

Among proteins that were specifically enriched on beads carrying unmodified K7 peptide, we found many proteins involved in early stages of transcription initiation (e.g., transcription factors Sox2 and Utf1, transcription initiation factor TFIID) or RNA processing (e. g., Serine/arginine-rich splicing factors Srsf1, Srsf5, Srsf7, Srsf 9; heterogeneous nuclear ribonucleoprotein A/B, subunits of H/ACA ribonucleoprotein complex, etc.). The second pull-down step with dimethylated K7 peptide did not reveal specific protein enrichment over mock background in two biological replicates. These results raise the possibility that CTDK7 methylation may act to prevent protein-protein interactions that occur in the earliest stages of transcription initiation (between transcription factors and RNAPII), which have to be disabled to allow CTD modifications that are compatible with productive elongation. Alternatively, it is possible that K7me1 and K7me2 act in concert with other CTD modifications, e.g. with S5p. In this light, more complex peptide pull-downs will be necessary where K7me2 presence/absence is combined with presence of S5p, S7p or other CTD modifications, to help understand what is the contribution of K7me2 to the ability of adjacent modifications to regulate co-transcriptional recruitment of machinery.

We believe that more complex and unbiased screens for CTDK7 binders will be necessary to identify factors that can recognize CTDK7me marks.

It could also be tested whether SET1, which is recruited by CTD S5P to methylate histone H3K4 at promoters, also methylates CTD K7me1/2.

To test the potential role of SET1 complexes as CTD-K7 methylases, we performed western blots for K7me1 and total RPB1 on protein extracts from control mouse ES cells and conditional knockout ES cells for Setd1a and Setd1b (extracts kindly provided by Francis Stewart; Biotechnology Center, Technische Universitaet Dresden; Figure 10). We do not observe consistent reduction in the levels of CTD- K7me1 in Setd1a-knockout cells and we observe only a minor depletion (15% max) in Setd1b-knockout cells (Figure 10). Normalization of CTD-K7me1 levels relative to total RPB1 was performed to correct for minor global changes in total RPB1 following SET knock out. Taken together, our results show that it is unlikely that SETD1A or SETD1B, individually, have major roles in K7 methylation.

We also tested conditional knockout cells for H3K4 methyltransferases Mll1 and Mll2 (extracts kindly provided by Francis Stewart; data not shown). We find that Mll1 or Mll2 knockout is accompanied by a large global increase in total RPB1 levels, making it difficult to assess the effect on CTD methylation. These results argue for an intricate interplay between histone and RNAPII modifications.

Mammalian cells have twelve H3K4 methyltransferases with redundant activity, and so it is possible that knockout of multiple enzymes may be needed to detect significant global changes in CTD-K7 methylation. Depletion of H3K4me3 levels may also have global effects on transcription that could influence the levels of CTD-K7 methylation independently of the direct effect of the methyltransferases. Alternative biochemical approaches such as CTD-K7 peptide pull down screens for methylases and/or isolation of CTD-K7 methylation activity from nuclear extracts might be needed for the identification of CTD-K7 methyltransferases. As these analyses are not conclusive at identifying K7me writers or readers we have not included these additional results in the manuscript.

Author response image 3.Screen of CTD-K7 methylase activity in conditional KO mouse ES cells for H3K4 methyltransferases.(**A**) Western blotting was performed using protein extracts from mouse ES cells with conditional knockout for H3K4 methylase Setd1a or Setd1b for total RPB1 and K7me1 in 2 biological replicates. Hypo- (IIa) and hyper-phosphorylated (II0) RPB1 forms are indicated. α-tubulin was used as loading control. (**B**) Signal intensities for total RPB1 and K7me1 (control and KO samples) were normalized to the corresponding α-tubulin signal to correct for variability in loading. Levels of K7me1 were normalized to the total amount of RPB1 and intensity relative to control cells is represented for the 2 replicates.**DOI:**
http://dx.doi.org/10.7554/eLife.11215.027